# Variance reduction for Random Coordinate Descent-Langevin Monte Carlo

**Zhiyan Ding**
Department of Mathematics
University of Wisconsin-Madison
Madison, WI 53706
zding49@math.wisc.edu

**Qin Li**
Department of Mathematics
University of Wisconsin-Madison
Madison, WI 53706
qinli@math.wisc.edu

## Abstract

Sampling from a log-concave distribution function is one core problem that has wide applications in Bayesian statistics and machine learning. While most gradient free methods have slow convergence rate, the Langevin Monte Carlo (LMC) that provides fast convergence requires the computation of gradients. In practice one uses finite-differencing approximations as surrogates, and the method is expensive in high-dimensions.

A natural strategy to reduce computational cost in each iteration is to utilize random gradient approximations, such as random coordinate descent (RCD) or simultaneous perturbation stochastic approximation (SPSA). We show by a counter-example that blindly applying RCD does not achieve the goal in the most general setting. The high variance induced by the randomness means a larger number of iterations are needed, and this balances out the saving in each iteration.

We then introduce a new variance reduction approach, termed Randomized Coordinates Averaging Descent (RCAD), and incorporate it with both overdamped and underdamped LMC. The methods are termed RCAD-O-LMC and RCAD-U-LMC respectively. The methods still sit in the random gradient approximation framework, and thus the computational cost in each iteration is low. However, by employing RCAD, the variance is reduced, so the methods converge within the same number of iterations as the classical overdamped and underdamped LMC [14, 12, 15]. This leads to a computational saving overall.

## 1 Introduction

Monte Carlo Sampling is one of the core problems in Bayesian statistics, data assimilation [59], and machine learning [1], with wide applications in atmospheric science [28], petroleum engineering [54], remote sensing [42] and epidemiology [43] in the form of inverse problems [49], volume computation [70], and bandit optimization [66].

Let $f(x)$ be a convex function that is $L$-gradient Lipschitz and $\mu$-strongly convex in $\mathbb{R}^d$. Define the target probability density function $p(x) \propto e^{-f}$, then $p(x)$ is a log-concave function. To sample from the probability distribution induced by $p(x)$ amounts to finding an $x \in \mathbb{R}^d$ (or a list of $\{x^i \in \mathbb{R}^d\}$) that can be regarded as i.i.d. (independent and identically distributed) drawn from the distribution.

There is vast literature on sampling, and proposed methods fall into a few different categories. Markov chain Monte Carlo (MCMC) [61] composes a big class of methods, including Metropolis-Hasting based MCMC (MH-MCMC) [52, 35], Gibbs samplers [32, 9], Hamiltonian Monte Carlo [55, 24], Langevin dynamics based methods [65] (including both the overdamped Langevin [58, 63, 13] and underdamped Langevin [11, 47] Monte Carlo), and some kind of combination (such as MALA) [63,

62, 25, 8]. Importance sampling and sequential Monte Carlo [34, 56, 17] framework and ensemble type methods [59, 31, 38, 18, 19, 20] are also popular.

Different MCMC methods are implemented differently, but they share the essence, that is to develop a Markovian transition kernel whose invariant measure is the target distribution, so that after many rounds of iteration, the invariant measure is achieved. If the design of the transition kernel does not involve $\nabla f$ or sense the local behavior of $f$, the convergence is slow [37, 36, 64, 51].

The Langevin Monte Carlo methods, both the overdamped or underdamped, can be viewed as special kinds of MCMC that involve the computation of $\nabla f$. The idea is to find stochastic differential equations (SDEs) whose equilibrium-in-time is the target distribution. These SDEs are typically driven by $\nabla f$, and the Overdamped or Underdamped Langevin Monte Carlo (O/U-LMC) can be viewed as the discrete-in-time (such as Euler-Maruyama discretization) version of the Langevin dynamics (SDEs). Since $\nabla f$ leads the dynamics, fast converge is expected [14, 12, 15].

However, $\nabla f$ is typically not available. In particular, if $f$ is obtained from inverse problems with an underlying governing differential equation describing the dynamics, as seen in the remote sensing and epidemiology examples above, the explicit formula for $\nabla f$ is unknown. When this happens, one usually needs to compute all partial derivatives, one by one, either by employing automatic differentiation [3], or by surrogating with the finite-difference approximations $\partial_i f \approx [f(x + \eta \mathbf{e}^i) - f(x - \eta \mathbf{e}^i)]/2\eta$ for every direction $\mathbf{e}^i$. This leads to a cost that is roughly $d$ times the number of required iterations. In high dimension, $d \gg 1$, the numerical cost is high. Therefore, how to sample with a small number of finite differencing approximations with a cost relaxed on $d$, becomes rather crucial.

There are methods proposed to achieve gradient-free property, such as Importance Sampling (IS), Ensemble Kalman methods, random walks methods, and various finite difference approximations to surrogate the gradient. However, IS [34, 22, 23] has high variance of the weight terms and it leads to wasteful sampling; ensemble Kalman methods [27, 5, 59, 31] usually require Gaussianity assumption [18, 19]; random walk methods such that Metropolized random walk (MRW) [51, 63, 64], Ball Walk [44, 26, 45] and the Hit-and-run algorithm [4, 40, 46] cannot guarantee fast convergence [69]; and to our best knowledge, modification of LMC with derivatives replaced by its finite difference approximation [50] or Kernel Hilbert space [68] are not yet equipped with theoretical non-asymptotic analysis.

## 1.1 Contribution

We work under the O/U-LMC framework, and we look for methods that produce i.i.d. samples with only a small number of gradient computation. To this end, the contribution of the paper is twofolded.

We first examine a natural strategy to reduce the cost by adopting randomized coordinate descent (RCD) [57, 72], a random directional gradient approximation. This method replaces $d$ finite difference approximations in $d$ directions, by 1 in a randomly selected direction. Presumably this reduces the cost in each iteration by $d$ folds, and hopefully the total cost. However, in this article we will show that this is not the case in the general setting. We will provide a counter-example: the high variance induced by the random direction selection process brings up the numerical error, and thus more iterations are needed to achieve the preset error tolerance. This in the end leads to no improvement in terms of the computational cost.

We then propose a variance reduction method to improve the application of RCD to LMC. We call the method Randomized Coordinates Averaging Descent Overdamped/Underdamped LMC (or RCAD-O/U-LMC). The methods start with a fully accurate gradient (up to a discretization error) in the first round of iteration, and in the subsequent iterations they only update the gradient evaluation in one randomly selected direction. Since the methods preserve some information about the gradient along the evolution, the variance is reduced. We prove the new methods converge as fast as the classical O/U-LMC [14, 12, 15], meaning the preset error tolerance is achieved in the same number of iterations. But since they require only 1 directional derivative per iteration instead of $d$, the overall cost is reduced. We summarize the advantage over the classical O-LMC and U-LMC in Table 1 (assuming computing the full gradient costs $d$ times of one partial derivative). The dependence on the conditioning of $f$ is omitted in the table, but will be discussed in detail in Section 5.

In some sense, the new methods share some similarity with SAGA [16], a modification of SAG (stochastic average gradient) [67]. These are two methods designed for reducing variance in the stochastic gradient descent (SGD) framework where the cost function $f$ has the form of $\sum_i f_i$. Similar approaches are also found in SG-MCMC (stochastic-gradient Markov chain Monte Carlo (SG-MCMC)) [47, 11, 30, 6, 7, 73, 10]. In their cases, variance reduction is introduced in the selection of $\nabla f_i$. In our case, the cost function $f$ is a simple convex function, but the gradient $\nabla f$ can be viewed as $\nabla f = \sum \partial_i f \mathbf{e}^i$ and the variance reduction is introduced in the selection of $\partial_i f \mathbf{e}^i$.

There are other variance reduction methods, such as SVRG [39] and CV-ULD [2, 10]. We leave the discussion to future research.

| Algorithm | Number of iterations | Number of $\partial f$ evaluations |
|---|---|---|
| O-LMC[14] | $\widetilde{O}\left(d/\epsilon\right)$ | $\widetilde{O}\left(d^2/\epsilon\right)$ |
| U-LMC[12, 15] | $\widetilde{O}\left(d^{1/2}/\epsilon\right)$ | $\widetilde{O}\left(d^{3/2}/\epsilon\right)$ |
| RCAD-O-LMC | $\widetilde{O}\left(d^{3/2}/\epsilon\right)$ | $\widetilde{O}\left(d^{3/2}/\epsilon\right)$ |
| RCAD-U-LMC | $\widetilde{O}\left(\max\{d^{4/3}/\epsilon^{2/3}, d^{1/2}/\epsilon\}\right)$ | $\widetilde{O}\left(\max\{d^{4/3}/\epsilon^{2/3}, d^{1/2}/\epsilon\}\right)$ |

Table 1: Number of iterations and directional derivative evaluations of $f(x)$ to achieve $\epsilon$-accuracy. $d$ is the dimension. $\widetilde{O}(f) = O(f \log f)$. If $g = O(f \log f)$, then $g \leq C f \log(f)$, where $C$ is a constant independent of $f$. For the overdamped cases, we assume the Lipschitz continuity for the hessian term. Without this assumption, RCAD-O-LMC still outperforms O-LMC, as will be discussed in Section 5.

## 1.2 Organization

In Section 2, we discuss the essential ingredients of our methods: the random coordinate descent (RCD) method, the overdamped and underdamped Langevin dynamics and the associated Monte Carlo methods (O-LMC and U-LMC). In Section 3, we unify the notations and assumptions used in our methods. In Section 4, we discuss the vanilla RCD applied to LMC and present a counter-example to show it is not effective if used blindly. In Section 5, we introduce our new methods RCAD-O/U-LMC and present the results on convergence and numerical cost. We demonstrate numerical evidence in Section 6. Proofs are rather technical and are all left to appendices.

## 2 Essential ingredients

### 2.1 Random coordinate descent (RCD)

When explicit formula for $\nabla f$ is not available, one needs to compute the partial derivatives for all directions. One straightforward way is to use finite difference: $\partial_i f(x) \approx \frac{f(x+\eta \mathbf{e}^i) - f(x-\eta \mathbf{e}^i)}{2\eta}$ where $\mathbf{e}^i$ is the $i$-th unit direction. Given enough smoothness, the introduced error is $O(\eta^2)$. For approximating the entire $\nabla f$, $d$ such finite differencing evaluations are required, and it is expensive in the high dimensional setting when $d \gg 1$. The cost is similarly bad if one uses automatic differentiation.

Ideally one can take one random direction and computes the derivative in that direction only, and hopefully this random directional derivative reveals some information of the entire gradient $\nabla f$. This approach is used in both RCD [72, 60, 57] and SPSA [33, 41]. Both methods, instead of calculating the full gradient, randomly pick one direction and use the directional derivative as a surrogate of $\nabla f$. More specifically, RCD computes the derivative in one random unit direction $\mathbf{e}^r$ and approximates:

$$\nabla f \approx d\left(\nabla f(x) \cdot \mathbf{e}^r\right) \mathbf{e}^r \approx d\frac{f(x+\eta \mathbf{e}^r) - f(x-\eta \mathbf{e}^r)}{2\eta} \mathbf{e}^r, \qquad (1)$$

where $r$ is randomly drawn from $1, 2, \cdots, d$ (see the distribution of drawing in [60]). This approximations is consistent in the expectation sense because

$$\mathbb{E}_r\left(d\left(\nabla f(x) \cdot \mathbf{e}^r\right) \mathbf{e}^r\right) = \nabla f(x).$$

Here $\mathbb{E}$ is to take expectation.

## 2.2 Overdamped Langevin dynamics and O-LMC

The O-LMC method is derived from the following Langevin dynamics:

$$\mathrm{d}X_t = -\nabla f(X_t)\,\mathrm{d}t + \sqrt{2}\,\mathrm{d}\mathcal{B}_t\,. \tag{2}$$

The SDE characterizes the trajectory of $X_t$. The forcing term $\nabla f(X_t)$ and the Brownian motion term $\mathrm{d}\mathcal{B}_t$ compete: the former drives $X_t$ to the minimum of $f$ and the latter provides small oscillations. The initial data $X_0$ is a random variable drawn from a given distribution induced by $q_0(x)$. Denote $q(x, t)$ the probability density function of $X_t$, it is a well-known result that $q(x, t)$ satisfies the following Fokker-Planck equation:

$$\partial_t q = \nabla \cdot (\nabla f q + \nabla q)\,, \quad \text{with} \quad q(x, 0) = q_0\,, \tag{3}$$

and furthermore, $q(x, t)$ converges to the target density function $p(x) = e^{-f}$ exponentially fast in time [48].

The overdamped Langevin Monte Carlo (O-LMC), as a sampling method, can be viewed as a discrete-in-time version of the SDE (2). A standard Euler-Maruyama method applied on the equation gives:

$$x^{m+1} = x^m - \nabla f(x^m)h + \sqrt{2h}\xi^m\,, \tag{4}$$

where $\xi^m$ is i.i.d. drawn from $\mathcal{N}(0, I_d)$ with $I_d$ being the identity matrix of size $d$. Since (4) approximates (2), the density of $x^m$, denoted as $p_m(x)$, converges to $p(x)$ as $m \to \infty$, up to a discretization error. It was proved in [14] that the convergence to $\epsilon$ is achieved within $\widetilde{O}(d/\epsilon)$ iterations if hessian of $f$ is Lipschitz. If hessian of $f$ is not Lipschitz, the number of iterations increases to $\widetilde{O}(d/\epsilon^2)$. In many real applications, the gradient of $f$ is not available and some approximation is used, introducing another layer of numerical error. In [14], the authors did discuss the effect of such error, but they assumed the error has bounded variance.

## 2.3 Underdamped Langevin dynamics and U-LMC

The underdamped Langevin dynamics [11] is characterized by the following SDE:

$$\begin{cases} \mathrm{d}X_t = V_t\,\mathrm{d}t \\ \mathrm{d}V_t = -2V_t\,\mathrm{d}t - \gamma\nabla f(X_t)\,\mathrm{d}t + \sqrt{4\gamma}\,\mathrm{d}\mathcal{B}_t \end{cases}, \tag{5}$$

where $\gamma > 0$ is a parameter to be tuned. Denote $q(x, v, t)$ the probability density function of $(X_t, V_t)$, then $q$ satisfies the Fokker-Planck equation

$$\partial_t q = \nabla \cdot \left( \begin{bmatrix} -v \\ 2v + \gamma\nabla f \end{bmatrix} q + \begin{bmatrix} 0 & 0 \\ 0 & 2\gamma \end{bmatrix} \nabla q \right)\,,$$

and under mild conditions, it converges to $p_2(x, v) = \exp(-(f(x) + |v|^2/2\gamma))$, making the marginal density function for $x$ the target $p(x)$ [71, 21].

The underdamped Langevin Monte Carlo algorithm, U-LMC, can be viewed as a numerical solver to (5). In each step, we sample new particles $(x^{m+1}, v^{m+1}) \sim (Z_x^{m+1}, Z_v^{m+1}) \in \mathbb{R}^{2d}$, where $(Z_x^{m+1}, Z_v^{m+1}) \in \mathbb{R}^{2d}$ is a Gaussian random vector determined by $(x^m, v^m)$ with the following expectation and covariance:

$$\begin{aligned}
\mathbb{E}Z_x^{m+1} &= x^m + \frac{1}{2}\left(1 - e^{-2h}\right)v^m - \frac{\gamma}{2}\left(h - \frac{1}{2}\left(1 - e^{-2h}\right)\right)\nabla f(x^m)\,, \\
\mathbb{E}Z_v^{m+1} &= v^m e^{-2h} - \frac{\gamma}{2}\left(1 - e^{-2h}\right)\nabla f(x^m)\,, \\
\mathrm{Cov}\left(Z_x^{m+1}\right) &= \gamma\left[h - \frac{3}{4} - \frac{1}{4}e^{-4h} + e^{-2h}\right]\cdot I_d\,,\ \mathrm{Cov}\left(Z_v^{m+1}\right) = \gamma\left[1 - e^{-4h}\right]\cdot I_d\,, \\
\mathrm{Cov}\left(Z_x^{m+1}, Z_v^{m+1}\right)) &= \frac{\gamma}{2}\left[1 + e^{-4h} - 2e^{-2h}\right]\cdot I_d\,.
\end{aligned} \tag{6}$$

We here used the notation $\mathbb{E}$ to denote the expectation, and $\mathrm{Cov}(a, b)$ to denote the covariance of $a$ and $b$. If $b = a$, we abbreviate it to $\mathrm{Cov}(a)$. The scheme can be interpreted as sampling from the

following dynamics in each time interval:

$$\begin{cases} \mathrm{X}_t = x^m + \displaystyle\int_0^t \mathrm{V}_s \, \mathrm{d}s \\ \mathrm{V}_t = v^m e^{-2t} - \dfrac{\gamma}{2}(1 - e^{-2t})\nabla f(x^m) + \sqrt{4\gamma}e^{-2t}\displaystyle\int_0^t e^{2s} \, \mathrm{d}\mathcal{B}_s \end{cases}.$$

U-LMC does demonstrate faster convergence rate [12, 15] than O-LMC. Without the assumption on the hessian of $f$ being Lipschitz, the number of iteration is $\widetilde{O}(\sqrt{d}/\epsilon)$ to achieve $\epsilon$ accuracy. The faster convergence on the discrete level could be explained by the better discretization solver instead of faster convergence of the underlying SDEs. Indeed, without the Lipschitz continuity on the hessian term, the discretizing of (5) produces $O(h^2)$ numerical error. In contrast, the discretization error of (4) is $O(h^{3/2})$. A third-order discretization was discussed for (5) in [53], further enhancing the numerical accuracy. Similar to O-LMC, the method needs to numerically approximate $\nabla f(x^m)$. This induces another layer of error, and also requires $d$ times of evaluation of $\partial f$.

## 3 Notations

### 3.1 Assumption

We make some standard assumptions on $f(x)$:

**Assumption 3.1.** *The function $f$ is $\mu$-strongly convex and has an L-Lipschitz gradient:*

- *Convex, meaning for any $x, x' \in \mathbb{R}^d$:*

$$f(x) - f(x') - \nabla f(x')^\top (x - x') \geq (\mu/2)|x - x'|^2 \,. \tag{7}$$

- *Gradient is Lipschitz, meaning for any $x, x' \in \mathbb{R}^d$:*

$$|\nabla f(x) - \nabla f(x')| \leq L|x - x'| \,. \tag{8}$$

If $f$ is second-order differentiable, these assumptions together mean $\mu I_d \preceq \mathcal{H}(f) \preceq L I_d$ where $\mathcal{H}(f)$ is the hessian of $f$. We also define condition number of $f(x)$ as

$$\kappa = L/\mu \geq 1 \,. \tag{9}$$

We will express our results in terms of $\kappa$ and $\mu$. Furthermore, for some results we assume Lipschitz condition of the hessian too:

**Assumption 3.2.** *The function $f$ is second-order differentiable and the hessian of $f$ is H-Lipschitz, meaning for any $x, x' \in \mathbb{R}^d$:*

$$\|\mathcal{H}(f)(x) - \mathcal{H}(f)(x')\|_2 \leq H|x - x'| \,. \tag{10}$$

### 3.2 Wasserstein distance

The Wasserstein distance is a classical quantity that evaluates the distance between two probability measures:

$$W_p(\mu, \nu) = \left( \inf_{(X,Y) \in C(\mu,\nu)} \mathbb{E}|X - Y|^p \right)^{1/p} \,,$$

where $C(\mu, \nu)$ is the set of distribution of $(X, Y) \in \mathbb{R}^{2d}$ whose marginal distributions, for $X$ and $Y$ respectively, are $\mu$ and $\nu$. These distributions are called the couplings of $\mu$ and $\nu$. Here $\mu$ and $\nu$ can be either probability measures themselves or the measures induced by probability density functions $\mu$ and $\nu$. In this paper we mainly study $W_2$.

## 4 Direct application of RCD in LMC, a negative result

We study if RCD can be blindly applied to U-LMC for reducing numerical complexity. This is to replace $\nabla f$ in the updating formula (4) for U-LMC by the random directional derivative surrogates (1). The resulting algorithms are presented as Algorithm 1 in Appendix A.1.

RCD was introduced in optimization. In [60], the authors show that despite RCD computes only $1$, instead of $d$ directional derivatives in each iteration, the number of iteration needed for achieving $\epsilon$-accuracy is $O(d/\epsilon)$, as compared to $O(1/\epsilon)$ when the full-gradient is used (suppose Lipschitz coefficient in each direction is at the same order with the total Lipschitz constant). The gain on the cost is mostly reflected by the conditioning of the objective function $f$. This means there are counter-examples for which RCD cannot save compared with ordinary gradient descent. We emphasize that there are of course also plenty examples for which RCD significantly outperforms when $f$ is special conditioning structures [60, 57, 72]. In this article we would like to investigate the general lower-bound situations.

The story is the same for sampling. There are examples that show directly applying the vanilla RCD to U-LMC fails to outperform the classical U-LMC. One example is the following: We assume

$$q_0(x,v) = \frac{1}{(4\pi)^{d/2}} \exp(-|x-\mathbf{u}|^2/2 - |v|^2/2)\,, \quad p_2(x,v) = \frac{1}{(2\pi)^{d/2}} \exp(-|x|^2/2 - |v|^2/2)\,,$$

where $\mathbf{u} \in \mathbb{R}^d$ satisfies $\mathbf{u}_i = 1/8$ for all $i$. Denote $\{(x^m, v^m)\}$ the sample computed through Algorithm 1 (underdamped) with stepsize $h$. Let $\eta$ be extremely small and the finite differencing error is negligible, and denote $q_m$ the probability density function of $(x^m, v^m)$, then we can show $W_2(q_m, p_2)$ cannot converge too fast.

**Theorem 4.1.** *For the example above, choose $\gamma = 1$, there exists uniform nonzero constant $C_1$ such that if $d, h$ satisfy*

$$d > 2, \quad h < \left\{ \frac{1}{100(1+C_1)}, \frac{1}{1440^2 d} \right\}\,,$$

*then*

$$W_m \geq \exp\left(-2mh\right) \frac{\sqrt{d}}{1024} + \frac{d^{3/2}h}{2304}\,, \tag{11}$$

*where $W_m = W_2(q_m^U, p_2)$, and $q_m^U(x,v)$ is the probability density function of $m$-th iteration of RCD-U-LMC.*

The proof is found in Section A.2. We note the second term in (11) is rather big. The smallness comes from $h$, the stepsize, and it needs be small enough to balance out the influence from $d^{3/2} \gg 1$. This puts strong restriction on $h$. Indeed, to have $\epsilon$-accuracy, $W(q_m, p_2) \leq \epsilon$, we need both terms smaller than $\epsilon$, and this term suggests that $h \leq \frac{2304\epsilon}{d^{3/2}}$ at least. And when combined with restriction from the first term, we arrive at the conclusion that at least $\widetilde{O}(d^{3/2}/\epsilon)$ iterations are needed, and thus $\widetilde{O}(d^{3/2}/\epsilon)$ finite differencing approximation are required. The $d$ dependence is $d^{3/2}$, and is exactly the same as that in U-LMC, meaning RCD-U-LMC brings no computational advantage over U-LMC in terms of the dependence on the dimension of the problem.

We emphasize that that large second term, as shown in the proof, especially in Section A.2 equation (A.11), is induced exactly due to the high variance in the gradient approximation. This triggers our investigation into variance reduction techniques.

## 5   Random direction approximation with variance reduction on O/U-LMC, two positive results

The direct application of RCD induces high variance and thus high error. It leads to many more rounds of iterations for convergence, gaining no numerical saving in the end. In this section we propose RCAD-O/U-LMC with RCAD reducing variance in the framework of RCD. We will prove that while the numerical cost per iteration is reduced by $d$-folds, the number of required iteration is mostly unchanged, and thus the total cost is reduced.

### 5.1   Algorithm

The key idea is to compute one accurate gradient at the very beginning in iteration No. 1, and to preserve this information along the iteration to prevent possible high variance. The algorithms for RCAD-O-LMC and RCAD-U-LMC are both presented in Algorithm 1, based on overdamped and underdamped Langevin dynamics respectively. Potentially the same strategy can be combined with SPSA, which we leave to future investigation.

In the methods, an accurate gradient (up to a finite-differencing error) is used in the first step, denoted by $g \approx \nabla f$, and in the subsequent iterations, only one directional derivative of $f$ gets computed and updated in $g$.

---

**Algorithm 1 Randomized Coordinate Averaging Decent O/U-LMC (RCAD-O/U-LMC)**

---

**Preparation:**
1. Input: $\eta$ (space stepsize); $h$ (time stepsize); $\gamma$ (parameter); $d$ (dimension); $M$ (stopping index) and $f(x)$.
2. Initial: *(overdamped)*: $x^0$ i.i.d. sampled from a initial distribution induced by $q_0(x)$ and calculate $g^0 \in \mathbb{R}^d$:

$$g_i^0 = \frac{f(x^0 + \eta \mathbf{e}^i) - f(x^0 - \eta \mathbf{e}^i)}{2\eta}, \quad 1 \leq i \leq d. \tag{12}$$

*(underdamped)*: $(x^0, v^0)$ i.i.d. sampled from a initial distribution induced by $q_0(x, v)$ and calculate $g^0 \in \mathbb{R}^d$ as in (12).
**Run: For** $m = 0, 1, \cdots M$
    1. Draw a random number $r^m$ uniformly from $1, 2, \cdots, d$.
    2. Calculate $g^{m+1}$ and flux $F^m \in \mathbb{R}^d$ by letting $g_i^{m+1} = g_i^m$ for $i \neq r^m$ and

$$g_{r^m}^{m+1} = \frac{f(x^m + \eta \mathbf{e}^{r^m}) - f(x^m - \eta \mathbf{e}^{r^m})}{2\eta}, \quad F^m = g^m + d\left(g^{m+1} - g^m\right). \tag{13}$$

    3. *(overdamped)*: Draw $\xi^m$ from $\mathcal{N}(0, I_d)$:

$$x^{m+1} = x^m - F^m h + \sqrt{2h}\xi^m. \tag{14}$$

*(underdamped)*: Sample $(x^{m+1}, v^{m+1}) \sim Z^{m+1} = (Z_x^{m+1}, Z_v^{m+1})$ where $Z^{m+1}$ is a Gaussian random variable with expectation and covariance defined in (6), replacing $\nabla f(x^m)$ by $F^m$.
**end**
**Output:** $\{x^m\}$.

---

## 5.2 Convergence and numerical cost analysis

We now discuss the convergence of RCAD-O-LMC and RCAD-U-LMC, and compare the results with the classical O-LMC and U-LMC methods [14, 12]. We emphasize that these two papers indeed discuss the numerical error in approximating the gradients, but they both require the variance of error being bounded, which is not the case here. One related work is [10], where the authors construct the Lyapunov function to study the convergence of SG-MCMC. Our proof for the convergence of RCAD-O-LMC is inspired by its technicalities. In [12, 10], a contraction map is used for U-LMC, but such map cannot be directly applied to our situation because the variance depends on the entire trajectory of samples. Furthermore, the history of the trajectory is reflected in each iteration, deeming the process to be non-Markovian. We need to re-engineer the iteration formula accordingly for tracing the error propagation.

### 5.2.1 Convergence for RCAD-O-LMC

For RCAD-O-LMC, we have the following theorem:

**Theorem 5.1.** *Suppose $f$ satisfies Assumption 3.1-3.2 and $h, \eta$ satisfy*

$$h < \frac{1}{3(1 + 9d)\kappa^2 \mu}, \quad \eta < h. \tag{15}$$

*Then $W_2(q_m^O, p)$, the Wasserstein distance between $q_m^O$, the probability density function of the sample $x^m$ derived from Algorithm 1 (overdamped), and $p$, the target density function, satisfies*

$$W_2(q_m^O, p) \leq \exp(-\mu h m/4)\sqrt{1 + 1/\kappa^2}W_2(q_0^O, p) + 2h\sqrt{d^3 C_1 + d^2 C_2}. \tag{16}$$

*Here $C_1 = 77\kappa^2\mu$, $C_2 = H^2/\mu^2 + 20\kappa^2 + \kappa^3\mu/d$.*

See proof in Appendix B. The theorem gives us the strategy of designing stopping criterion: to achieve $\epsilon$-accuracy, meaning to have $W_2(q_m^O, p) \leq \epsilon$, we can choose to set both terms in (16) less than $\epsilon/2$, and it leads to:

$$h \leq \min\left\{\frac{1}{3(1+9d)\kappa^2\mu}, \frac{\epsilon}{4d^{3/2}\sqrt{C_1 + C_2/d}}\right\}$$

and

$$M \geq \frac{4}{h\mu}\log\left(\frac{2\sqrt{1+1/\kappa^2}W_2(q_0, p)}{\epsilon}\right).$$

This means the cost, also the number of $\partial f$ evaluations, is $\widetilde{O}(d^{3/2}/\epsilon)$.

Note that the theorem here requires both Assumptions 3.1 and 3.2. We can relax the second assumption. If so, the numerical cost of degrades to $\widetilde{O}(\max\{d^{3/2}/\epsilon, d/\epsilon^2\})$, whereas the cost of O-LMC is $\widetilde{O}(d^2/\epsilon^2)$. Our strategy still outperforms. The proof is the same, and we omit it from the paper.

### 5.2.2 Convergence for RCAD-U-LMC

For RCAD-U-LMC, we have the following theorem.

**Theorem 5.2.** *Assume $f(x)$ satisfies Assumption 3.1, and set $\gamma = 1/L$, then there exists a uniformly constant $D > 0$ such that if $h, \eta$ satisfy*

$$h \leq \min\left\{\frac{1}{100(1+D)\kappa}, \frac{1}{1648\kappa d}\right\}, \quad \eta < h^3, \tag{17}$$

*then $W_2(q_m^U, p_2)$, the Wasserstein distance between the distribution of the sample $(x^m, v^m)$, derived from Algorithm 1 (underdamped), and distribution induced by $p_2$ (whose marginal density in $x$ is $p$) decays as:*

$$\begin{aligned} W_2(q_m^U, p_2) \leq &4\sqrt{2}\exp(-hm/(8\kappa))W_2(q_0^U, p_2) \\ &+ 600\sqrt{h^3 d^4/\mu} + 200\sqrt{\kappa h^2 d/\mu} + 350\sqrt{\kappa h^5 d^2}. \end{aligned} \tag{18}$$

See proof in Appendix C. To achieve $\epsilon$-accuracy, meaning to have $W_2(q_m^U, p_2) \leq \epsilon$, we can choose all terms in (18) less than $\epsilon/4$. This gives:

$$h \leq \min\left\{\frac{\epsilon^{2/3}\mu^{1/3}}{(2400)^{2/3}d^{4/3}}, \frac{\epsilon\mu^{1/2}}{800\kappa^{1/2}d^{1/2}}, \frac{\epsilon^{2/5}}{(1400)^{2/5}\kappa^{1/5}d^{2/5}}, \frac{1}{(1+D)\kappa}, \frac{1}{1648\kappa d}\right\}$$

and thus the stopping index needs to be:

$$M \geq \frac{8\kappa}{h}\log\left(\frac{16\sqrt{2}W_2(q_0^U, p_2)}{\epsilon}\right).$$

This means $\widetilde{O}\left(\max\left\{d^{4/3}/\epsilon^{2/3}, d^{1/2}/\epsilon\right\}\right)$ evaluations of $\partial f$.

## 6 Numerical result

We demonstrate numerical evidence in this section. We first note that it is extremely difficult to compute the Wasserstein distance between two probability measures in high dimensional problems, especially when they are represented by a number of samples. The numerical result below evaluates a weaker measure:

$$\text{Error} = \left|\frac{1}{N}\sum_{i=1}^N \phi(x^{M,i}) - \mathbb{E}_p(\phi)\right|, \tag{19}$$

where $\phi$ is the test function. $\{x^{M,i}\}_{i=1}^N$ are $N$ different samples iterate till $M$-th step, and $p$ is the target distribution.

In the first example, our target distribution is $\mathcal{N}(0, I_d)$ with $d = 1000$, and in the second example we use

$$p(x) \propto \exp\left(-\sum_{i=1}^{d} \frac{|x_i - 2|^2}{2}\right) + \exp\left(-\sum_{i=1}^{d} \frac{|x_i + 2|^2}{2}\right).$$

The initial distributions, for the overdamped and underdamped situations respectively, are $\mathcal{N}(\mathbf{0.5}, I_d)$ and $\mathcal{N}(\mathbf{0.5}, I_{2d})$ in both exampes. We run both RCD-O/U-LMC and RCAD-O/U-LMC using $N = 5 \times 10^5$ particles and test MSE error with $\phi(x) = |x_1|^2$ in both examples. In Figure 1 and Figure 2 respectively we show the error with respect to different stepsizes. In all the computation, $M$ is big enough. The improvement of adding variance reduction technique is obvious in both examples.

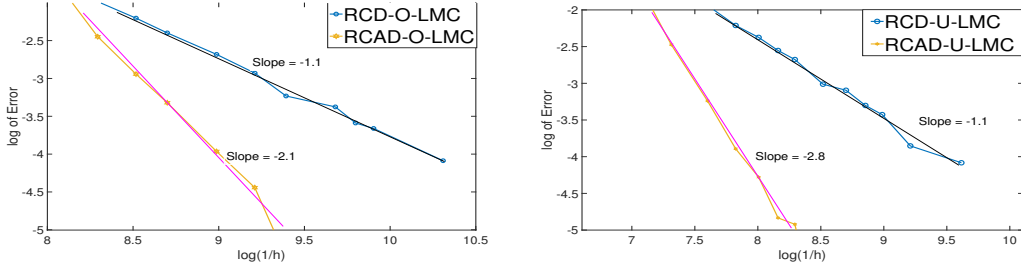

Figure 1: Example 1. Decay of Error of O-LMC (left) and U-LMC (right) with and without RCAD.

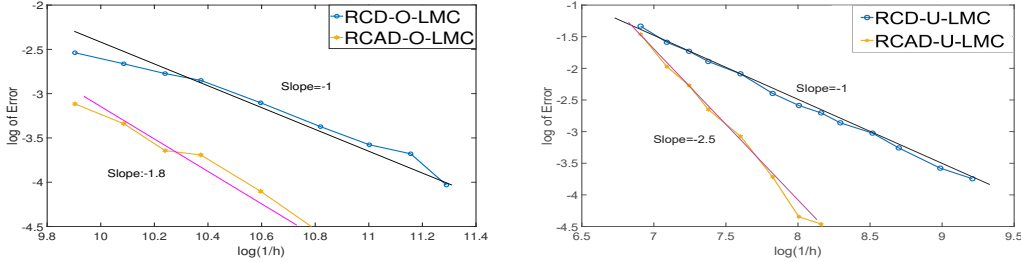

Figure 2: Example 2. Decay of Error of O-LMC (left) and U-LMC (right) with and without RCAD.

## 7   Conclusion and future work

To our best knowledge, this is the first work that discusses both the negative and positive aspects of applying random gradient approximation, mainly RCD type, to LMC, in both overdamped and underdamped situations without and with variance reduction. Without variance reduction we show the RCD-LMC has the same numerical cost as the classical LMC, and with variance reduction, the numerical cost is reduced in both overdamped and underdamped cases.

There are a few future directions that we would like to pursue. 1. Our method, in its current version, is blind to the structure of $f$. The only assumptions are reflected on the Lipschitz bounds. In [60, 57, 29] the authors, in studying optimization problems, propose to choose random directions according to the Lipschitz constant in each direction. The idea could potentially be incorporated in our framework to enhance the sampling strategy. 2. Our algorithms are designed based on reducing variance in the RCD framework. Potentially one can also apply variance reduction methods to improve SPSA-LMC. There are also other variance reduction methods that one could explore.

## 8   Broader Impact

The result provides theoretical guarantee to the application of random coordinate descent to Langevin Monte Carlo, when variance reduction technique is used to reduce the cost. It has potential application

to inverse problems emerging from atmospheric science, remote sensing, and epidemiology. This work does not present any foreseeable societal consequence.

## Acknowledgments and Disclosure of Funding

Both authors acknowledge generous support from NSF-DMS 1750488, NSF-TRIPODS 1740707, Wisconsin Data Science Initiative, and Wisconsin Alumni Research Foundation.

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
