[Supplementary Material]

# A Algorithms and Results of RCD-LMC

## A.1 Algorithm

We apply RCD as surrogates of the gradient in O/U-LMC. This amounts to replacing the gradient terms in (4) using the approximation (1). The new methods are presented in Algorithm 1, termed RCD-O/U-LMC.

---

**Algorithm 1 RCD-overdamped(underdamped) Langevin Monte Carlo**

---

**Preparation:**
1. Input: $\eta$ (space step); $h$ (time step); $\gamma$ (parameter); $d$ (dimension); $M$ (stopping index) and $f(x)$.
2. Initial: *(overdamped)*: $x^0$ i.i.d. sampled from a initial distribution induced by $q_0(x)$.
  *(underdamped)*: $(x^0, v^0)$ i.i.d. sampled from the initial distribution induced by $q_0(x, v)$.
**Run: For** $m = 0, 1, \cdots M$
  1. Finite difference: calculate flux approximation by RCD:

$$F^m = d\frac{f(x^m + \eta\mathbf{e}^r) - f(x^m - \eta\mathbf{e}^r)}{2\eta}\mathbf{e}^r \qquad (A.1)$$

with $r$ uniformly drawn from $1, \cdots, d$.
  2. *(overdamped)*: Draw $\xi^m$ from $\mathcal{N}(0, I_d)$:

$$x^{m+1} = x^m - F^m h + \sqrt{2h}\xi^m. \qquad (A.2)$$

*(underdamped)*: Sample $(x^{m+1}, v^{m+1}) \sim Z^{m+1} = (Z_x^{m+1}, Z_v^{m+1})$ where $Z^{m+1}$ is a Gaussian random variable with expectation and covariance defined in (6), replacing $\nabla f(x^m)$ by $F^m$.
**end**
**Output:** $\{x^m\}$.

---

## A.2 A counter-example

In this section, we prove Theorem 4.1.

Fisrt, we define $w^m = x^m + v^m$, and denote $u_m(x, w)$ the probability density of $(x^m, w^m)$ and $u^*(x, w)$ the probability density of $(x, w)$ if $(x, v = w - x)$ is distributed according to density function $p_2$. From [2], we have:

$$|x^m - x|^2 + |v^m - v|^2 \leq 4(|x^m - x|^2 + |w^m - w|^2) \leq 16(|x^m - x|^2 + |v^m - v|^2), \quad (A.3)$$

and thus

$$W_2^2(q_m^U, p_2) \leq 4W_2^2(u_m, u^*) \leq 16W_2^2(q_m^U, p_2). \qquad (A.4)$$

*Proof of Theorem 4.1.* Throughout the proof, we drop the superscript "$U$" to have a concise notation. According to (A.4), it suffices to find a lower bound for $W_2^2(u_m, u^*)$. We first notice

$$W_2(u_m, u^*) \geq \sqrt{\int\int |w|^2 u_m(x, w)\,\mathrm{d}w\,\mathrm{d}x} - \sqrt{\int\int |w|^2 u^*(x, w)\,\mathrm{d}w\,\mathrm{d}x}$$

$$= \sqrt{\int\int |w|^2 u_m(x, w)\,\mathrm{d}w\,\mathrm{d}x} - \sqrt{2d} = \sqrt{\mathbb{E}|w^m|^2} - \sqrt{2d} \qquad (A.5)$$

$$= \frac{\mathbb{E}|w^m|^2 - 2d}{\sqrt{\mathbb{E}|w^m|^2} + \sqrt{2d}},$$

where $\mathbb{E}$ takes all randomness into account. This implies to prove (11), it suffices to find a lower bound for second moment of $w^m$. Indeed, in the end, we will show that

$$W_2(u_m, u^*) \geq (1 - 2h)^m \frac{\sqrt{d}}{512} + \frac{d^{3/2}h}{1152} \qquad (A.6)$$

and thus

$$W_2(q_m, p_2) \geq (1 - 2h)^m \frac{\sqrt{d}}{1024} + \frac{d^{3/2}h}{2304}$$

proving the statement of the theorem since $(1 - 2h)^m \geq \exp(-2mh)$. To show (A.6), we first note, by direct calculation:

$$W_2(q_0, p_2) = \sqrt{d}/8, \quad \mathbb{E}|\omega^0|^2 = \frac{129d}{64}, \quad (A.7)$$

then we divide the proof into several steps:

- **First step:** *Priori estimation*

  According to (A.7), use convergence result of Algorithm 1 ([4] Theorem 4.1), we have for any $m \geq 0$

  $$W_2(q_m, p_2) \leq \frac{\sqrt{d}}{2} + \frac{\sqrt{d}}{6} = \frac{2\sqrt{d}}{3},$$

  Similar to (A.5), we have

  $$W_2(q_m, p_2) \geq |\sqrt{\mathbb{E}|x^m|^2} - \sqrt{d}|, \quad W_2(q_m, p) \geq |\sqrt{\mathbb{E}|v^m|^2} - \sqrt{d}|$$

  which implies

  $$\frac{\sqrt{d}}{3} \leq \sqrt{\mathbb{E}|x^m|^2} \leq \frac{5\sqrt{d}}{3}, \quad \frac{\sqrt{d}}{3} \leq \sqrt{\mathbb{E}|v^m|^2} \leq \frac{5\sqrt{d}}{3} \quad (A.8)$$

  for any $m \geq 0$.

  Finally, use (A.8), we can obtain

  $$\sqrt{\mathbb{E}|\omega^m|^2} \leq \sqrt{\mathbb{E}|x^m|^2} + \sqrt{\mathbb{E}|v^m|^2} \leq 4\sqrt{d}. \quad (A.9)$$

- **Second step:** *Iteration formula of $\mathbb{E}|w^m|^2$.*

  By the special structure of $p$, we can calculate the second moment explicitly. Since $f(x)$ can be written as

  $$f(x) = \sum_{i=1}^{d} \frac{|x_i|^2}{2},$$

  in each step of RCD-U-LMC, according to Algorithm 1, for each $m \geq 0$ and $1 \leq i \leq d$, we have

  $$\mathbb{E}\left(x_i^{m+1}|(x^m, v^m, r^m)\right) = x_i^m + \frac{1 - e^{-2h}}{2}v_i^m - \left(\frac{h}{2} - \frac{1 - e^{-2h}}{4}\right)(x_i^m - E_i^m),$$

  $$\mathbb{E}\left(v_i^{m+1}|(x^m, v^m, r^m)\right) = v_i^m e^{-2h} - \frac{1 - e^{-2h}}{2}(x_i^m - E_i^m),$$

  $$\mathbb{E}\left(w_i^{m+1}|(x^m, v^m, r^m)\right) = \frac{1 + e^{-2h}}{2}w_i^m + \frac{1 - e^{-2h}}{2}E_i^m - \left(\frac{h}{2} - \frac{1 - e^{-2h}}{4}\right)(x_i^m - E_i^m),$$

  $$\mathrm{Var}\left(x_i^{m+1}|(x^m, v^m, r^m)\right) = h - \frac{3}{4} - \frac{1}{4}e^{-4h} + e^{-2h},$$

  $$\mathrm{Var}\left(v_i^{m+1}|(x^m, v^m, r^m)\right) = 1 - e^{-4h},$$

  $$\mathrm{Cov}\left((x_i^{m+1}, v_i^{m+1})|(x^m, v^m, r^m)\right) = \frac{1}{2}\left[1 + e^{-4h} - 2e^{-2h}\right], \quad (A.10)$$

  where $E^m \in \mathbb{R}^d$ is a random variable defined as

  $$E_i^m = x_i^m - dx_i^m \mathbf{e}_i^{r_m}$$

  and satisfies

  $$\mathbb{E}_{r_m}(E_i^m) = 0, \quad \mathbb{E}_{r_m}|E_i^m|^2 = (d - 1)|x_i^m|^2 \quad (A.11)$$

  for each $1 \leq i \leq d$. Furthermore,

  $$\mathbb{E}\langle w_i^m, E_i^m \rangle = \mathbb{E}\langle x_i^m, E_i^m \rangle = 0. \quad (A.12)$$

Now, since $h \leq \frac{1}{880}$, we can replace $e^{-2h}$ and $e^{-4h}$ by their Taylor expansion:

$$e^{-2h} = 1 - 2h + 2h^2 + D_1 h^3, \quad e^{-4h} = 1 - 4h + 8h^2 + D_2 h^3, \qquad \text{(A.13)}$$

where $D_1, D_2$ are negative constants depends on $h$ and satisfy

$$|D_1| < 10, \quad |D_2| < 100.$$

Plug (A.13) into (A.10), we have

$$\mathbb{E}\left(w_i^{m+1}|(x^m, w^m, r^m)\right) = \left(1 - h + h^2 + \frac{D_1 h^3}{2}\right) w_i^m - \left(\frac{h^2}{2} + \frac{D_1 h^3}{4}\right) x_i^m$$
$$+ \left(h - \frac{h^2}{2} - \frac{D_1 h^3}{4}\right) E_i^m,$$

$$\text{Var}\left(x_i^{m+1}|(x^m, v^m, r^m)\right) = \left(D_1 - \frac{D_2}{4}\right) h^3,$$

$$\text{Var}\left(v_i^{m+1}|(x^m, v^m, r^m)\right) = 4h - 8h^2 - D_2 h^3,$$

$$\text{Cov}\left((x_i^{m+1}, v_i^{m+1})|(x^m, v^m, r^m)\right) = 2h^2 + \frac{(D_2 - 2D_1) h^3}{2}.$$

$$\text{(A.14)}$$

The last three equalities in (A.14) implies

$$\text{Var}\left(w_i^{m+1}|(x^m, v^m, r^m)\right) = \text{Var}\left(x_i^{m+1}|(x^m, v^m, r^m)\right) + \text{Var}\left(v_i^{m+1}|(x^m, v^m, r^m)\right)$$
$$+ 2\text{Cov}\left(x_i^{m+1}, v_i^{m+1})|(x^m, v^m, r^m)\right)$$
$$= 4h - 4h^2 - \left(D_1 + \frac{D_2}{4}\right) h^3.$$

Then, we can calculate the iteration formula for $\mathbb{E}|x_i^{m+1}|^2$ and $\mathbb{E}|\omega_i^{m+1}|^2$:

$$\mathbb{E}|\omega_i^{m+1}|^2$$
$$= \mathbb{E}_{x^m, w^m, r^m}\left(|\omega_i^{m+1}|^2 \big| (x^m, v^m, r^m)\right)$$
$$= \mathbb{E}_{x^m, w^m, r^m}\left(\left|\mathbb{E}\left(w_i^{m+1}|(x^m, w^m, r^m)\right)\right|^2 + \text{Var}\left(w_i^{m+1}|(x^m, v^m, r^m)\right)\right)$$
$$= \left(1 - h + h^2 + \frac{D_1 h^3}{2}\right)^2 \mathbb{E}|w_i^m|^2 + (d-1)\left(h - h^2 - \frac{D_1 h^3}{2}\right)^2 \mathbb{E}|x_i^m|^2$$
$$+ \left(\frac{h^2}{2} + \frac{D_1 h^3}{4}\right)^2 \mathbb{E}|x_i^m|^2 - 2\left(1 - h + h^2 + \frac{D_1 h^3}{2}\right)\left(\frac{h^2}{2} + \frac{D_1 h^3}{4}\right) \mathbb{E}\langle w_i^m, x_i^m \rangle$$
$$+ 4h - 4h^2 - \left(D_1 + \frac{D_2}{4}\right) h^3,$$

where we use (A.11) and (A.12).

Sum them up with $i$, we finally obtain an iteration formula for $\mathbb{E}|w^m|^2$:

$$\mathbb{E}|\omega^{m+1}|^2$$
$$= \left(1 - h + h^2 + \frac{D_1 h^3}{2}\right)^2 \mathbb{E}|w^m|^2 + (d-1)\left(h - h^2 - \frac{D_1 h^3}{2}\right)^2 \mathbb{E}|x^m|^2$$
$$+ \left(\frac{h^2}{2} + \frac{D_1 h^3}{4}\right)^2 \mathbb{E}|x^m|^2 - 2\left(1 - h + h^2 + \frac{D_1 h^3}{2}\right)\left(\frac{h^2}{2} + \frac{D_1 h^3}{4}\right) \mathbb{E}\langle w^m, x^m \rangle$$
$$+ 4dh - 4dh^2 - d\left(D_1 + \frac{D_2}{4}\right) h^3.$$

$$\text{(A.15)}$$

- **Third step:** *Lower bound for $W_2(u_m, u^*)$* Use (A.9), since $D_1 < 0$, $h < \frac{1}{100} < \frac{1}{|D_1|}$ and $d > 1872$, we have

$$\frac{h^2}{2} \leq h^2 + \frac{D_1 h^3}{2} \leq h^2, \quad h - h^2 - \frac{D_1 h^3}{2} \geq h - h^2 \geq \frac{h}{2},$$

which implies

$$1 - h + h^2 + \frac{D_1 h^3}{2} \geq 1 - h + h^2/2, \quad (d-1)\left(h - h^2 - \frac{D_1 h^3}{2}\right)^2 \mathbb{E}|x^m|^2 \geq \frac{d^2 h^2}{72}. \tag{A.16}$$

and

$$\left(\frac{h^2}{2} + \frac{D_1 h^3}{4}\right) \mathbb{E}\langle w^m, x^m\rangle \leq \frac{h^2}{2}\left(\mathbb{E}|w^m|^2 \mathbb{E}|x^m|^2\right)^{1/2} \leq 4dh^2. \tag{A.17}$$

For the last line of (A.15), since $h \leq \frac{1}{880} < \frac{1}{|D_1| + |D_2|/4}$, we have

$$4dh - 4dh^2 - d\left(D_1 + \frac{D_2}{4}\right)h^3 \geq 4dh - 5dh^2. \tag{A.18}$$

Plug (A.16),(A.17),(A.18) into (A.15), we have

$$\mathbb{E}|\omega^{m+1}|^2 \geq \left(1 - h + h^2/2\right)^2 \mathbb{E}|w^m|^2 + 4dh + d^2 h^2/72 - 13dh^2. \tag{A.19}$$

Note that $\left(1 - h + h^2/2\right)^2 \geq 1 - 2h$, and since $d > 1872$, we have $\frac{d^2 h^2}{144} \geq 13dh^2$. Use (A.19) iteratively and combine with (A.7), we finally have:

$$\begin{aligned}
\mathbb{E}|\omega^m|^2 &\geq \frac{129\left(1 - 2h\right)^m d}{64} + \left(1 - (1 - 2h)^m\right)\left[2d + d^2 h/288\right] \\
&= (1 - 2h)^m\left[\frac{d}{64} - \frac{d^2 h}{288}\right] + 2d + \frac{d^2 h}{288} \\
&\geq (1 - 2h)^m \frac{d}{128} + 2d + \frac{d^2 h}{288},
\end{aligned} \tag{A.20}$$

Plug (A.20) into (A.5), we further have

$$\begin{aligned}
W_2(u_m, u^*) &\geq \frac{(1 - 2h)^m \frac{d}{128} + 2d + \frac{d^2 h}{288} - 2d}{\sqrt{(1 - 2h)^m \frac{d}{128} + 2d + \frac{d^2 h}{288}} + \sqrt{2d}} \\
&\geq \frac{(1 - 2h)^m \frac{d}{128} + \frac{d^2 h}{288}}{4\sqrt{d}} \\
&\geq (1 - 2h)^m \frac{\sqrt{d}}{512} + \frac{d^{3/2} h}{1152},
\end{aligned}$$

where we use small enough $h$ in the second inequality to bound the $d^2 h$ term by $d$ in the denominator. This proves (A.6).

$\square$

# B  Proof of convergence of RCAD-O-LMC (Theorem 5.1)

In this section we provide the detailed proof for Theorem 5.1.

Before diving into details, we quickly summarize the proving strategy. Recall that the target distribution $p$ is merely the equilibrium of the SDE (2). This means, if a particle prepared at the initial stage is drawn from $p$, then following the dynamics of SDE (2), the distribution of this particle will continue to be $p$. In the analysis below, we call the trajectory of this particle $y_t$, and the sequence generated by this particle evaluated at discrete time $y^m$. Essentially we evaluate how quickly $x^m$ converges to $y^m$ as $m$ increases. In particular, we call $\Delta^m = x^m - y^m$ and will derive an iteration formula that shows the convergence of $\Delta^m$.

In evaluating $\Delta^m$, there are three kinds of error that get involved:

1. discretization error in $\eta$: this can be made as small as possible. $\eta$ is a spatial stepsize parameter and can be made as small as we wish. The finite differencing accuracy is second order and thus the produced error is at the order of $\mathcal{O}(\eta^2)$. By making $\eta$ small, we make this part of error negligible;

2. discretization error in $h$: this amounts to controlling the discretization error of the SDE (2). To handle this part of error we employ the estimates in [3];

3. random coordinate selection process error: this is to measure, at each iteration, how big can $\nabla f(x^m) - F^m$ be. According to the way $F^m$ is defined, it is straightforward to show that the expectation of this error is always 0, but the variance $\mathbb{E}|\nabla f(x^m) - F^m|^2$ can be big, and this is the main reason for the direct application of RCD on LMC to fail [4]. The variance reduction technique discussed in this paper is exactly to reduce the size of this term.

In a way, see details in (B.11), we can derive the iteration formula, ignoring the discretization error in $\eta$,

$$
\begin{aligned}
\Delta^{m+1} =& y^{m+1} - x^{m+1} = \Delta^m + (y^{m+1} - y^m) - (x^{m+1} - x^m) \\
=& \Delta^m - h\left(\nabla f(y^m) - \nabla f(x^m)\right) - \int_{mh}^{(m+1)h} \left(\nabla f(y_s) - \nabla f(y^m)\right) \mathrm{d}s \\
& - h(\nabla f(x^m) - F^m)\,.
\end{aligned}
$$

The second term on the right hand side, by using the Lipschitz continuity, will provide $-Lh\Delta^m$, and it produces desirable property when combined with the first $\Delta^m$. The third term encodes the discretization error in $h$, and was shown to be small in [3]. The last term is the error that comes from the random coordinate selection process. We discuss it in details in Section D. We note that this term, due to the complicated random coordinate selection process, cannot be uniquely controlled by $\Delta^m$, but rather, the entire trajectory of the selection process has to enter. To overcome that, we define a the Lyapunov function that singles out $\partial_{r^m} f(x^m) - \partial_{r^m} f(y^m)$, a term that has the $r^m$ dependence. The boundedness of this term, and the iteration formula for $\Delta^m$ combined give the decay of the Lyapunov function. See definition in (B.8).

Now we prove the theorem in details. After a lengthy definition of all notations, we will present Lemma B.1 and Lemma B.2. They are to bound, iteratively $\Delta^m$ and $\partial_{r^m} f(x^m) - \partial_{r^m} f(y^m)$ term respectively. The proof of the theorem then follows by combining the two lemmas to control the Lyapunov function.

As presented in the main text, the first step of RCAD-O-LMC uses the finite differencing approximation for every direction, namely, setting $g^0 \in \mathbb{R}^d$ to be:

$$
g_i^0 = \frac{f(x^0 + \eta \mathbf{e}^i) - f(x^0 - \eta \mathbf{e}^i)}{2\eta}, \quad i = 1, 2, \cdots, d\,.
$$

In the following iterations, one random direction is selected for the updating,

$$
g_{r_m}^{m+1} = \frac{f(x^m + \eta \mathbf{e}^{r_m}) - f(x^m - \eta \mathbf{e}^{r_m})}{2\eta}
$$

with other directions untouched: $g_i^{m+1} = g_i^m$ for all $i \neq r_m$. Define:

$$
F^m = g^m + d\left(g^{m+1} - g^m\right)\,,
$$

then the updating formula is:

$$
x^{m+1} = x^m - F^m h + \sqrt{2h}\xi^m\,, \tag{B.1}
$$

where $h$ is the time stepsize, and $\xi^m$ i.i.d. drawn from $\mathcal{N}(0, I_d)$. Denote

$$
E^m = \nabla f(x^m) - F^m\,, \tag{B.2}
$$

then this updating formula (B.1) writes to:

$$
x^{m+1} = x^m - \nabla f(x^m)h + E^m h + \sqrt{2h}\xi^m\,. \tag{B.3}
$$

This is the formula we use for the analysis under Assumptions 3.1 and 3.2.

To show the theorem, we let $y_0$ be a random vector drawn from target distribution induced by $p$ such that $W_2^2(q_0^O, p) = \mathbb{E}|x^0 - y^0|^2$, and set

$$y_t = y_0 - \int_0^t \nabla f(y_s)\,\mathrm{d}s + \sqrt{2}\int_0^t \mathrm{d}\mathcal{B}_s\,, \tag{B.4}$$

where we construct the Brownian motion that always satisfies

$$B_{h(m+1)} - B_{hm} = \sqrt{h}\xi^m\,. \tag{B.5}$$

Then $y_t$ is drawn from target distribution as well. On the discrete level, let $y^m = y_{mh}$, then:

$$y^{m+1} = y^m - \int_{mh}^{(m+1)h} \nabla f(y_s)\,\mathrm{d}s + \sqrt{2h}\xi^m\,.$$

Noting

$$W_2^2(q_m^O, p) \leq \mathbb{E}|x^m - y^m|^2\,,$$

where $\mathbb{E}$ takes all randomness into account. We now essentially need to show the difference between (B.1) and (B.4), also see [1].

As for a preparation, we now define an a set of auxiliary gradients.

- $\widetilde{g}^0$ is the true derivative used at the initial step:

$$\widetilde{g}^0 = \nabla f(x^0)\,, \tag{B.6}$$

- $\widetilde{g}^{m+1}$ is the continuous version of $g^{m+1}$:

$$\widetilde{g}^{m+1}_{r_m} = \partial_{r_m} f(x^m) \quad \text{and} \quad \widetilde{g}^{m+1}_i = \widetilde{g}^m_i \quad \text{if} \quad i \neq r_m\,, \tag{B.7}$$

- $\widetilde{F}^m$ is the continuous version of $F^m$:

$$\widetilde{F}^m = \widetilde{g}^m + d\left(\widetilde{g}^{m+1} - \widetilde{g}^m\right)\,.$$

- Define $\beta^m$ using (B.6),(B.7) with the same $r_m$ but replacing $x^m$ with $y^m$:

$$\beta^0 = \nabla f(y^0)$$

and

$$\beta^{m+1}_{r_m} = \partial_{r_m} f(y^m) \quad \text{and} \quad \beta^{m+1}_i = \beta^m_i \quad \text{if} \quad i \neq r_m\,.$$

Indeed in the later proof we will give an upper bound for the following Lyapunov function:

$$T^m = T_1^m + c_p T_2^m = \mathbb{E}|y^m - x^m|^2 + c_p\mathbb{E}|\widetilde{g}^m - \beta^m|^2\,. \tag{B.8}$$

where $c_p$ will be carefully chosen later.

We further define

$$\widetilde{E}^m = \nabla f(x^m) - \widetilde{F}^m = E^m + F^m - \widetilde{F}^m\,,$$

this leads to $E^m = \widetilde{E}^m - F^m + \widetilde{F}^m$. The properties of $\widetilde{E}^m$ will be discussed in Appendix D. To quantify $F^m - \widetilde{F}^m$ is straightforward: it can be bounded using mean-value theorem. Since:

$$|\widetilde{g}^0_i - g^0_i|^2 = \left|\frac{f(x^m + \eta \mathbf{e}^i) - f(x^m - \eta \mathbf{e}^i) - 2\eta\partial_i f(x^m)}{2\eta}\right|^2 \leq \left|\frac{(\partial_i f(z) - \partial_i f(x^m))2\eta}{2\eta}\right|^2 \leq L^2\eta^2$$

where $z \in \mathbb{R}^d$ is a point between $x^m + \eta \mathbf{e}^i$ and we use the fact that $\nabla f$ is $L$-Lipschitz. Similarly, for all $m$:

$$|\widetilde{g}^m - g^m|^2 \leq L^2\eta^2 d\,,$$

we have:

$$\left|\widetilde{F}^m - F^m\right|^2 \leq 2|\widetilde{g}^m - g^m|^2 + 2d^2|\widetilde{g}^{m+1}_{r_m} - g^{m+1}_{r_m}|^2 < 2L^2\eta^2 d + 8L^2\eta^2 d^2\,. \quad . \tag{B.9}$$

Now we present the iteration formula for $T_1^{m+1}, T_2^{m+1}$, in Lemma B.1 and Lemma B.2 respectively:

**Lemma B.1.** *Under conditions of Theorem 5.1, for any $a > 0$, we can upper bound $T_1^m$:*

$$T_1^{m+1} \le (1+a)AT_1^m + (1+a)BT_2^m + (1+a)h^3C + \left(1 + \frac{1}{a}\right)h^4D \qquad \text{(B.10)}$$

*where*

$$A = 1 - 2\mu h + 3(1+3d)L^2h^2\,, \quad B = 9h^2d\,,$$

$$C = 2L^2d + 72L^2d^3\left[\frac{hL^2d}{\mu} + 1\right]\,, \quad D = (H^2 + 16L^2)d^2 + (L^3 + 4L^2)d\,.$$

Note that for the proof to proceed, one at least needs the coefficient $(1+a)A < 1$. This can be made possible only if $a$ is small enough. For small $a$, the $h^4D$ term is magnified, but it may not matter as $h^4$ serves as a high order error so the term is negligible so long as $a \gg h^4$.

*Proof.* Define $\Delta^m = y^m - x^m$, we first divide $\Delta^{m+1}$ into several parts:

$$
\begin{aligned}
\Delta^{m+1} =& \Delta^m + (y^{m+1} - y^m) - (x^{m+1} - x^m)\\
=& \Delta^m + \left(-\int_{mh}^{(m+1)h} \nabla f(y_s)\,\mathrm{d}s + \sqrt{2h}\xi_m\right) - \left(-\int_{mh}^{(m+1)h} F^m\,\mathrm{d}s + \sqrt{2h}\xi_m\right)\\
=& \Delta^m - \left(\int_{mh}^{(m+1)h} (\nabla f(y_s) - F^m)\,\mathrm{d}s\right)\\
=& \Delta^m - \left(\int_{mh}^{(m+1)h} (\nabla f(y_s) - \nabla f(y^m) + \nabla f(y^m) - \nabla f(x^m) + \nabla f(x^m) - F^m)\,\mathrm{d}s\right),\\
=& \Delta^m - h\left(\nabla f(y^m) - \nabla f(x^m)\right) - \int_{mh}^{(m+1)h} (\nabla f(y_s) - \nabla f(y^m))\,\mathrm{d}s\\
& - h(\nabla f(x^m) - F^m)\\
=& \Delta^m - hU^m - (V^m + h\Phi^m) - hE^m\\
=& \Delta^m - (V^m + h(\widetilde{F}^m - F^m)) - h(U^m + \Phi^m + \widetilde{E}^m)
\end{aligned}
$$
$$\text{(B.11)}$$

where we set[1]

$$U^m = \nabla f(y^m) - \nabla f(x^m)\,,$$

$$V^m = \int_{mh}^{(m+1)h}\left(\nabla f(y_s) - \nabla f(y^m) - \sqrt{2}\int_{mh}^{s} \mathcal{H}(f)(y_r)\,\mathrm{d}B_r\right)\mathrm{d}s\,,$$

$$\Phi^m = \frac{\sqrt{2}}{h}\int_{mh}^{(m+1)h}\int_{mh}^{s} \mathcal{H}(f)(y_r)\,\mathrm{d}B_r\,\mathrm{d}s\,.$$

Upon getting equation (B.11) it is time to analyze each term and hopefully derive an induction inequality that states $\mathbb{E}|\Delta^{m+1}|^2 \approx c\mathbb{E}|\Delta^m|^2 + d$ with $c < 1$ and $d$ being of high order in $\eta$ and $h$, some parameters we can tune. Indeed the $\Delta^m$ term is what we would like to preserve, and the $U^m$ term depends on $\Delta^m$ with a Lipschitz coefficient. The opposite signs of these two terms essentially indicate that $c$ can be made $< 1$. The $V^m + h\Phi^m$ completely depends on the one-time step error. In some sense, it is close to the forward Euler error obtained in one timestep. The $\widetilde{E}^m$ term is the most crucial term and the only term that reflects the error introduced by the algorithm in one time step. By choosing the right discretization in the algorithm to approximate $\nabla f$, one could expect this term to be small. We leave the analysis of this term to Appendix D, and focus on how the other terms interact here.

We first control last two terms in the last line of (B.11). According to Lemma 6 of [3], we first have

$$\mathbb{E}|V^m|^2 \le \frac{h^4}{2}\left(H^2d^2 + L^3d\right)\,, \quad \mathbb{E}|\Phi^m|^2 \le \frac{2L^2hd}{3}\,, \qquad \text{(B.12)}$$

and thus:

$$\mathbb{E}|V^m + h(\widetilde{F}^m - F^m)|^2 \leq 2\left(\mathbb{E}|V^m|^2 + h^2\mathbb{E}|\widetilde{F}^m - F^m)|^2\right)$$

$$\leq h^4\left(H^2d^2 + L^3d\right) + 2h^2\left(2L^2\eta^2d + 8L^2\eta^2d^2\right) \tag{B.13}$$

$$\leq (H^2 + 16L^2)h^4d^2 + (L^3 + 4L^2)h^4d = h^4D\,,$$

where we use (B.9) and (B.12) in the second inequality and the condition of $h$ and $\eta$ in (15) in last inequality. We also have:

$$\mathbb{E}|U^m + \Phi^m + \widetilde{E}^m|^2$$

$$\leq 3\mathbb{E}|U^m|^2 + 3\mathbb{E}|\Phi^m|^2 + 3\mathbb{E}|\widetilde{E}^m|^2\,, \tag{B.14}$$

$$\leq 3L^2T_1^m + 2L^2hd + 9dL^2T_1^m + 9dT_2^m + 72hL^2d^3\left[\frac{hL^2d}{\mu} + 1\right]\,.$$

where we used the Lipschitz continuity of $f$ for controlling $U^m$, (B.12) for $\Phi^m$, and Appendix D for $\widetilde{E}^m$.

We then handle the cross terms. For example, due to the independence, (D.1), and the convexity, we have:

$$\mathbb{E}\langle\Delta^m, \Phi^m\rangle = 0\,, \quad \mathbb{E}\left\langle\Delta^m, \widetilde{E}^m\right\rangle = 0\,, \quad \langle\Delta^m, U^m\rangle \geq \mu|\Delta^m|^2\,, \tag{B.15}$$

this means the cross term between first and the third term in the last line (B.11) leads to $-2\mu h\mathbb{E}|\Delta^m|^2$. The cross term produced by the first and the last term, however can be hard to control, mostly because $\mathbb{E}\langle\Delta^m, V^m\rangle$ is unknown. We now employ Young's inequality, meaning, for any $a > 0$:

$$T_1^{m+1} = \mathbb{E}|\Delta^{m+1}|^2$$

$$\leq (1+a)\mathbb{E}|\Delta^{m+1} + V^m + h(\widetilde{F}^m - F^m)|^2 + \left(1 + \frac{1}{a}\right)\mathbb{E}|V^m + h(\widetilde{F}^m - F^m)|^2\,. \tag{B.16}$$

While the second term is already investigated in (B.13), the first term of (B.16), according to (B.11) becomes:

$$\mathbb{E}|\Delta^{m+1} + V^m + h(\widetilde{F}^m - F^m)|^2 = \mathbb{E}|\Delta^m - h(U^m + \Phi^m + \widetilde{E}^m)|^2$$

$$= \mathbb{E}|\Delta^m|^2 - 2h\mathbb{E}\left\langle\Delta^m, U^m + \Phi^m + \widetilde{E}^m\right\rangle$$

$$+ h^2\mathbb{E}|U^m + \Phi^m + \widetilde{E}^m|^2 \tag{B.17}$$

$$\leq (1 - 2\mu h)\mathbb{E}|\Delta^m|^2 + h^2\mathbb{E}|U^m + \Phi^m + \widetilde{E}^m|^2$$

where we used (B.15). Plug(B.14) into (B.17), we have have, using the definition of the coefficients $A, B, C$:

$$\mathbb{E}|\Delta^m - h(U^m + \Phi^m + \widetilde{E}^m)|^2 \leq AT_1^m + Ch^3 + BT_2^m\,, \tag{B.18}$$

and plug it together with (B.13) in (B.16) to conclude (B.10). $\qquad\square$

**Lemma B.2.** *Under conditions of Theorem 5.1, we have the upper bound for $T_2^{m+1}$:*

$$T_2^{m+1} \leq \tilde{A}T_1^m + \tilde{B}T_2^m \tag{B.19}$$

*where $\tilde{A} = \frac{L^2}{d}$ and $\tilde{B} = 1 - 1/d$.*

Note that the coefficient $\tilde{B}$ is automatically $< 1$ and the gap $1/d$ is independent of $h$ and $\eta$. This gives us some room to tune the parameters.

*Proof.* We now expand $\mathbb{E}\left|\beta_i^{m+1} - \widetilde{g}_i^{m+1}\right|^2$:

$$\mathbb{E}_{r_m}\left|\beta_i^{m+1} - \widetilde{g}_i^{m+1}\right|^2 = \mathbb{E}_{r_m}\left[\left|\beta_i^{m+1} - \widetilde{g}_i^{m+1}\right|^2 - |\beta_i^m - \widetilde{g}_i^m|^2\right] + |\beta_i^m - \widetilde{g}_i^m|^2$$

$$= \frac{1}{d}\left[|\partial_i f(y^m) - \partial_i f(x^m)|^2 - |\beta_i^m - \widetilde{g}_i^m|^2\right] + |\beta_i^m - \widetilde{g}_i^m|^2\,.$$

$$= \left(1 - \frac{1}{d}\right)|\beta_i^m - \widetilde{g}_i^m|^2 + \frac{1}{d}|\partial_i f(y^m) - \partial_i f(x^m)|^2$$

Therefore, we have

$$\mathbb{E}\left|\beta^{m+1} - \widetilde{g}^{m+1}\right|^2 = \left(1 - \frac{1}{d}\right)\mathbb{E}\sum_{i=1}^{d}|\beta_i^m - \widetilde{g}_i^m|^2 + \frac{1}{d}\mathbb{E}|\nabla f(y^m) - \nabla f(x^m)|^2$$

$$\leq \left(1 - \frac{1}{d}\right)\mathbb{E}\left|\beta^m - \widetilde{g}^m\right|^2 + \frac{L^2}{d}\mathbb{E}|\Delta^m|^2$$

(B.20)

$\square$

Now, we are ready to prove Theorem 5.1 by adjusting $a$ and $c_p$.

*Proof of Theorem 5.1.* Plug (B.10) and (B.19) into (B.8), we have

$$T^{m+1} \leq \left((1+a)A + c_p\tilde{A}\right)T_1^m + \left(\frac{(1+a)B}{c_p} + \tilde{B}\right)c_pT_2^m$$

$$+ (1+a)h^3C + \left(1 + \frac{1}{a}\right)h^4D.$$

(B.21)

To show the proof amounts to choosing proper $c_p$ and $a$. Note that according to the definitions, $A \sim 1 - \mu h$, $\tilde{A} \sim 1/d$, $B \sim h^2$ and $\tilde{B} \sim 1 - 1/d$, this suggests $c_p \sim h^2$ to cancel out the order in $B$, and in the end we have estimates of the form:

$$(1+a)A + c_p\tilde{A} = 1 - O(h)\,, \quad \frac{(1+a)B}{c_p} + \tilde{B} = 1 - O(h)\,.$$

Indeed, let us choose

$$c_p = 18(1+a)h^2d^2\,,$$

so that

$$(1+a)A + c_p\tilde{A} = (1+a)(1 - 2\mu h + 3(1+9d)L^2h^2)\,, \quad \text{and} \quad \frac{(1+a)B}{c_p} + \tilde{B} = 1 - \frac{1}{2d}\,.$$

Since $h$ satisfies (15), this relaxes them to

$$(1+a)A + c_p\tilde{A} \leq (1+a)(1 - \mu h)\,, \quad \text{and} \quad \frac{(1+a)B}{c_p} + \tilde{B} = 1 - \frac{1}{2d} \leq 1 - \frac{\mu h}{2}\,.$$

Setting $a = \frac{\mu h/2}{1 - \mu h}$ so that

$$(1+a)(1 - \mu h) = 1 - \frac{\mu h}{2}\,, \quad \text{and} \quad 1 + 1/a \leq 2/\mu h\,,$$

and this finally leads to

$$T^{m+1} \leq (1 - \mu h/2)T_1^m + (1 - \mu h/2)c_pT_2^m + 2h^3C + \frac{2}{\mu h}h^4D$$

$$\leq (1 - \mu h/2)T^m + 2\left(h^3C + h^3D/\mu\right)\,.$$

(B.22)

Noting

$$W_2^2(q_m^O, p) \leq T^m$$

and

$$T^0 = \mathbb{E}|y^0 - x^0|^2 + c_p\mathbb{E}|g^0 - \beta^0|^2 = \mathbb{E}|y^0 - x^0|^2 + c_p\mathbb{E}|\nabla f(x^0) - \nabla f(y^0)|^2$$

$$\leq (1 + c_pL^2)\mathbb{E}|y^0 - x^0|^2 \leq (1 + \mu^2/L^2)W_2^2(q_0^O, p) \leq (1 + 1/\kappa^2)W_2^2(q_0^O, p)\,,$$

where we use $c_pL^2 \leq 36h^2L^2d^2$ and $hLd < \mu/(27L)$, by iteration, we finally have

$$W_2^2(q_m^O, p) \leq \exp(-\mu hm/2)(1 + 1/\kappa^2)W_2^2(q_0^O, p) + 4\left(h^2C/\mu + h^2D/\mu^2\right)\,.$$

(B.23)

The proof is concluded considering

$$C/\mu \leq d^3\left(2L^2/(d^2\mu) + 75L^2/\mu\right) \leq 77d^3\kappa^2\mu\,,$$

$$D/\mu^2 \leq d^2(H^2/\mu^2 + 20\kappa^2 + \kappa^3\mu/d)\,.$$

$\square$

# C  Proof of convergence of RCAD-U-LMC (Theorem 5.2)

Recall the definitions:

- $E^m$: $E^m = \nabla f(x^m) - F^m$
- $\widetilde{g}^0$ : $\widetilde{g}^0 = \nabla f(x^0)$
- $\widetilde{g}^{m+1}$: $\widetilde{g}^{m+1}_{r_m} = \partial_{r_m} f(x^m)$ and $\widetilde{g}^{m+1}_i = \widetilde{g}^m_i$ if $i \neq r_m$,
- $\widetilde{F}^m$: $\widetilde{F}^m = \widetilde{g}^m + d\left(\widetilde{g}^{m+1} - \widetilde{g}^m\right)$
- $\widetilde{E}^m$: $\widetilde{E}^m = \nabla f(x^m) - \widetilde{F}^m = E^m + F^m - \widetilde{F}^m$

Similarly, we also have

$$|\widetilde{g}^m - g^m|^2 \leq L^2\eta^2 d, \quad \left|\widetilde{F}^m - F^m\right|^2 = \left|\widetilde{E}^m - E^m\right|^2 \leq 2L^2\eta^2 d + 8L^2\eta^2 d^2 . \tag{C.1}$$

According to the algorithm, RCAD-U-LMC can be seen as drawing $(x^0, v^0)$ from distribution induced by $q_0^U$, and update $(x^m, v^m)$ using the following coupled SDEs:

$$\begin{cases} V_t = v^m e^{-2(t-mh)} - \gamma \displaystyle\int_{mh}^t e^{-2(t-s)}\, ds F^m + \sqrt{4\gamma}e^{-2(t-mh)}\displaystyle\int_{mh}^t e^{2s} d\mathcal{B}_s \\[2mm] X_t = x^m + \displaystyle\int_{mh}^t V_s ds \end{cases}, \tag{C.2}$$

where $\mathcal{B}_s$ is the Brownian motion and $(x^{m+1}, v^{m+1}) = (X_{(m+1)h}, V_{(m+1)h})$.

We then define $w^m = x^m + v^m$, and denote $u_m(x, w)$ the probability density of $(x^m, w^m)$ and $u^*(x, w)$ the probability density of $(x, w)$ if $(x, v = w - x)$ is distributed according to density function $p_2$. One main reason to change $(x, v)$ to $(x, w)$ is that in [2], the authors showed that the map $(x_0, w_0) \to (x_t, w_t)$ induced from (5) is a contracting map for for $t$. From [2], we also have:

$$|x^m - x|^2 + |v^m - v|^2 \leq 4(|x^m - x|^2 + |w^m - w|^2) \leq 16(|x^m - x|^2 + |v^m - v|^2) \tag{C.3}$$

and

$$W_2^2(q_m^U, p_2) \leq 4W_2^2(u_m, u^*) \leq 16W_2^2(q_m^U, p_2) . \tag{C.4}$$

Similar to RCAD-O-LMC, define another trajectory of sampling by setting $(\widetilde{x}^0, \widetilde{v}^0)$ to be drawn from the distribution induced by $p_2$, and that $\widetilde{x}^m = \widetilde{X}_{hm}$, $\widetilde{v}^m = \widetilde{V}_{hm}$, $\widetilde{w}^m = \widetilde{x}^m + \widetilde{v}^m$ are samples from $\left(\widetilde{X}_t, \widetilde{V}_t\right)$ that satisfy

$$\begin{cases} \widetilde{V}_t = \widetilde{v}_0 e^{-2t} - \gamma \displaystyle\int_0^t e^{-2(t-s)}\nabla f\left(\widetilde{X}_s\right)\, ds + \sqrt{4\gamma}e^{-2t}\displaystyle\int_0^t e^{2s} d\mathcal{B}_s \\[2mm] \widetilde{X}_t = \widetilde{x}_0 + \displaystyle\int_0^t \widetilde{V}_s ds \end{cases}, \tag{C.5}$$

with the same Brownian motion as before. This leads to

$$\begin{cases} \widetilde{v}^{m+1} = \widetilde{v}^m e^{-2h} - \gamma \displaystyle\int_{mh}^{(m+1)h} e^{-2((m+1)h-s)}\nabla f(\widetilde{X}_s)\, ds + \sqrt{4\gamma}e^{-2h}\displaystyle\int_{mh}^{(m+1)h} e^{2s} d\mathcal{B}_s \\[2mm] \widetilde{x}^{m+1} = \widetilde{x}^m + \displaystyle\int_{mh}^{(m+1)h} \widetilde{V}_s ds \end{cases}. \tag{C.6}$$

Clearly $\left(\widetilde{X}_t, \widetilde{V}_t\right)$ can be seen as drawn from target distribution for all $t$, and initially we can pick $(\widetilde{x}^0, \widetilde{v}^0)$ such that

$$W_2^2(q_0^U, p_2) = \mathbb{E}\left(|x^0 - \widetilde{x}^0|^2 + |v^0 - \widetilde{v}^0|^2\right), \quad \text{and} \quad W_2^2(u_0, u^*) = \mathbb{E}\left(|x^0 - \widetilde{x}^0|^2 + |w^0 - \widetilde{w}^0|^2\right) .$$

We then also define $\beta^m$

$$\beta^0 = \nabla f(\widetilde{x}^0)$$

and

$$\beta_{r_m}^{m+1} = \partial_{r_m} f(\widetilde{x}^m) \quad \text{and} \quad \beta_i^{m+1} = \beta_i^m \quad \text{if} \quad i \neq r_m \,,$$

We will be showing the decay of the following Lyapunov function:

$$T^m \triangleq T_1^m + c_p T_2^m = \mathbb{E}\left(|\widetilde{x}^m - x^m|^2 + |\widetilde{w}^m - w^m|^2\right) + c_p \mathbb{E}|\widetilde{g}^m - \beta^m|^2 \,, \quad \text{(C.7)}$$

where $c_p$ will be carefully chosen later.

The following lemma gives bounds for $T_1^{m+1}, T_2^{m+1}$ using $T_1^m, T_2^m$, and the proof of the theorem amounts to selecting the correct $c_p$.

**Lemma C.1.** *Under conditions of Theorem 5.2, we have*

$$T_1^{m+1} < D_1 T_1^m + D_2 T_2^m + D_3 \,, \quad \text{(C.8)}$$

$$T_2^{m+1} \leq \frac{L^2}{d} T_1^m + \left(1 - \frac{1}{d}\right) T_2^m \,, \quad \text{(C.9)}$$

where

$$D_1 = 1 - h/(2\kappa) + 244h^2 d, \ D_2 = 84\gamma^2 h^2 d, \ D_3 = 672\gamma h^4 d^4 + 30h^3 d/\mu + 260h^6 d^2 \,.$$

*Proof.* The proof for bounding $T_2^m$ is the same as the one in Appendix B Lemma B.2 and is omit from here. We only prove the first inequality.

- **Step 1:** We firstly define $|\Delta^m|^2 = |\widetilde{w}^m - w^m|^2 + |\widetilde{x}^m - x^m|^2$, and compare (C.2) and (C.6) for:

$$
\begin{aligned}
|\Delta^{m+1}|^2 = \Bigg| & (\widetilde{v}^m - v^m)e^{-2h} + (\widetilde{x}^m - x^m) + \int_{mh}^{(m+1)h} \widetilde{V}_s - V_s \, ds \\
& - \gamma \int_{mh}^{(m+1)h} e^{-2((m+1)h-s)} \left[\nabla f\left(\widetilde{X}_s\right) - \nabla f(x^m)\right] \, ds \\
& + \gamma \int_{mh}^{(m+1)h} e^{-2((m+1)h-s)} E^m \, ds \Bigg|^2 \\
& + \left| (\widetilde{x}^m - x^m) + \int_{mh}^{(m+1)h} \widetilde{V}_s - V_s \, ds \right|^2 \\
= & |J_1^m|^2 + |J_2^m|^2 = \left| J_1^{r,m} + J_1^{E,m} \right|^2 + |J_2^m|^2 \,,
\end{aligned}
$$

where we denote

$$
\begin{aligned}
J_1^{r,m} = & (\widetilde{v}^m - v^m)e^{-2h} + (\widetilde{x}^m - x^m) + \int_{mh}^{(m+1)h} \widetilde{V}_s - V_s \, ds \\
& - \gamma \int_{mh}^{(m+1)h} e^{-2((m+1)h-s)} \left[\nabla f\left(\widetilde{X}_s\right) - \nabla f(x^m)\right] \, ds
\end{aligned}
$$

and

$$J_1^{E,m} = \gamma \int_{mh}^{(m+1)h} e^{-2((m+1)h-s)} E^m \,.$$

To control $J_1^m$, we realize that $J_1^{E,m}$ term, produced by $E^m$, is not perpendicular to the rest of the terms, namely $J_1^{r,m}$, and it will lead to a lot of cross terms. We thus replace it by $J_1^{\widetilde{E},m}$ induced by $\widetilde{E}^m$. This allows us to eliminate all cross terms. Since $E^m - \widetilde{E}^m$ is small, such replacement brings only small perturbation. In particular, with Young's inequality:

$$
\begin{aligned}
\mathbb{E}\,|J_1^m|^2 \leq & (1+h^2)\mathbb{E}\left|J_1^m + J_1^{\widetilde{E},m} - J_1^{E,m}\right|^2 + (1+1/h^2)\mathbb{E}\left|J_1^{\widetilde{E},m} - J_1^{E,m}\right|^2 \\
\leq & (1+h^2)\mathbb{E}\left|J_1^m + J_1^{\widetilde{E},m} - J_1^{E,m}\right|^2 + \gamma^2(h^2+1)(2L^2\eta^2 d + 8L^2\eta^2 d^2)
\end{aligned} \quad \text{(C.10)}
$$

where we use the smallness of $E^m - \tilde{E}^m$ in (C.1). The first term of (C.10) can be separated into three terms:

$$\mathbb{E}\left|\mathrm{J}_1^m + \mathrm{J}_1^{\tilde{E},m} - \mathrm{J}_1^{E,m}\right|^2 = \mathbb{E}\left|\mathrm{J}_1^{r,m} + \mathrm{J}_1^{\tilde{E},m}\right|^2$$

$$=\mathbb{E}\left|\mathrm{J}_1^{r,m}\right|^2 + \mathbb{E}\left|\mathrm{J}_1^{\tilde{E},m}\right|^2 + 2\mathbb{E}\left\langle \mathrm{J}_1^{r,m}, \mathrm{J}_1^{\tilde{E},m}\right\rangle$$

Firstly note that

$$\mathbb{E}\left|\mathrm{J}_1^{\tilde{E},m}\right|^2 \leq \gamma^2 h^2 \mathbb{E}\left|\tilde{E}^m\right|^2 .$$

And to bound the third term, note that

$$\mathbb{E}\left\langle \mathrm{J}_1^{r,m}, \mathrm{J}_1^{\tilde{E},m}\right\rangle = \mathbb{E}\left\langle \int_{mh}^{(m+1)h} \tilde{\mathrm{V}}_s - \mathrm{V}_s \, \mathrm{d}s , \mathrm{J}_1^{\tilde{E},m}\right\rangle$$

due to the fact that

$$\mathbb{E}\langle A, \tilde{E}^m\rangle = \mathbb{E}\langle A, \mathbb{E}_{r_m}\tilde{E}^m\rangle = 0 \tag{C.11}$$

for all $A$ that has no $r_m$ dependence. To further bound this term, we plug in the definition and have:

$$2\mathbb{E}\left\langle \int_{mh}^{(m+1)h} \tilde{\mathrm{V}}_s - \mathrm{V}_s \, \mathrm{d}s, \gamma \int_{mh}^{(m+1)h} e^{-2((m+1)h-s)}\,\mathrm{d}s\tilde{E}^m\right\rangle$$

$$= -2\mathbb{E}\left\langle \int_{mh}^{(m+1)h} \mathrm{V}_s \, \mathrm{d}s, \gamma \int_{mh}^{(m+1)h} e^{-2((m+1)h-s)}\,\mathrm{d}s\tilde{E}^m\right\rangle$$

$$=2\mathbb{E}\left\langle \gamma \int_{mh}^{(m+1)h}\int_{mh}^{s} e^{-2(s-t)}\,\mathrm{d}t\,\mathrm{d}sE^m, \gamma \int_{mh}^{(m+1)h} e^{-2((m+1)h-s)}\,\mathrm{d}s\tilde{E}^m\right\rangle,$$

$$\leq \gamma^2 h^3 (3\mathbb{E}\left|\tilde{E}^m\right|^2 + 4L^2\eta^2 d + 16L^2\eta^2 d^2)$$

where we used (C.11) again in the first and second equalities and

$$\mathbb{E}\left\langle E^m, \tilde{E}^m\right\rangle \leq 3\mathbb{E}|\tilde{E}^m|^2 + 2\mathbb{E}\left|\tilde{E}^m - E^m\right|^2$$

together with (C.1) in the last inequality.

In conclusion, we have

$$T_1^{m+1} = \mathbb{E}\left|\Delta^{m+1}\right|^2 \leq (1+h^2)\mathbb{E}\left|\mathrm{J}_1^{r,m}\right|^2 + |\mathrm{J}_2^m|^2 + \gamma^2(h^2+1)(2L^2\eta^2 d + 8L^2\eta^2 d^2)$$
$$+ (1+h^2)\left(\gamma^2 h^2 \mathbb{E}\left|\tilde{E}^m\right|^2 + \gamma^2 h^3(3\mathbb{E}\left|\tilde{E}^m\right|^2 + 4L^2\eta^2 d + 16L^2\eta^2 d^2)\right).\tag{C.12}$$

Using $\gamma L = 1, h < 1, \eta < h^3$, we have

$$T_1^{m+1} = \mathbb{E}\left|\Delta^{m+1}\right|^2 \leq (1+h^2)\mathbb{E}\left|\mathrm{J}_1^{r,m}\right|^2 + |\mathrm{J}_2^m|^2 + 2\gamma^2(h^2+3h^3)\mathbb{E}\left|\tilde{E}^m\right|^2.$$
$$+ 60h^6 d^2 \tag{C.13}$$

- **Step 2:** Now, we study first two terms in (C.13). We try to bound $(1+h^2)\mathbb{E}\left|\mathrm{J}_1^{r,m}\right|^2 + |\mathrm{J}_2^m|^2$ using $T_1^m$ and $\mathbb{E}|\tilde{E}^m|^2$. We first try to separate out $(x^m, \tilde{x}^m, v^m, \tilde{v}^m)$ from $\mathrm{J}_1^{r,m}$ and $\mathrm{J}_2^m$. Denote

$$A^m = (\tilde{v}^m - v^m)(h + e^{-2h}) + (\tilde{x}^m - x^m)$$
$$- \gamma \int_{mh}^{(m+1)h} e^{-2((m+1)h-s)}\left[\nabla f(\tilde{x}^m) - \nabla f(x^m)\right]\,\mathrm{d}s, \tag{C.14}$$

$$B^m = \int_{mh}^{(m+1)h} \tilde{\mathrm{V}}_s - \mathrm{V}_s - (\tilde{v}^m - v^m)\,\mathrm{d}s$$
$$- \gamma \int_{mh}^{(m+1)h} e^{-2((m+1)h-s)}\left[\nabla f\left(\tilde{\mathrm{X}}_s\right) - \nabla f(\tilde{x}^m)\right]\,\mathrm{d}s, \tag{C.15}$$

$$C^m = (\widetilde{x}^m - x^m) + \int_{mh}^{(m+1)h} \widetilde{v}^m - v^m \, \mathrm{d}s = (\widetilde{x}^m - x^m) + h(\widetilde{v}^m - v^m)\,, \tag{C.16}$$

$$D^m = \int_{mh}^{(m+1)h} \widetilde{V}_s - V_s - (\widetilde{v}^m - v^m) \, \mathrm{d}s\,, \tag{C.17}$$

then we have

$$J_1^{r,m} = A^m + B^m\,, \quad J_2^m = C^m + D^m\,.$$

By Young's inequality, we have

$$
\begin{aligned}
(1+h^2)\mathbb{E}|J_1^{r,m}|^2 + \mathbb{E}|J_2^m|^2 &= (1+h^2)\mathbb{E}|A^m + B^m|^2 + \mathbb{E}|C^m + D^m|^2 \\
&\leq (1+a)\left((1+h^2)\mathbb{E}|A^m|^2 + \mathbb{E}|C^m|^2\right) \\
&\quad + (1+1/a)((1+h^2)\mathbb{E}|B^m|^2 + \mathbb{E}|D^m|^2)\,,
\end{aligned}
\tag{C.18}
$$

where $a > 0$ will be carefully chosen later. Now, the first term of (C.18) only contains information from previous step, using $f$ is strongly convex, we can bound it using $|\Delta^m|^2$ (showed in Lemma E.3). To bound the second term, we need to consider difference between $x, v$ at $t_{m+1}$ and $t_m$, which can be bounded by $|\Delta^m|^2$ and $|E^m|^2$ (showed in Lemma E.2).

According to Lemma E.2-E.3, we first have

$$
\begin{aligned}
&(1+h^2)\mathbb{E}|J_1^{r,m}|^2 + \mathbb{E}|J_2^m|^2 \\
&\leq (1+a)\left[1 - h/\kappa + Dh^2\right]T_1^m \\
&\quad + (1+1/a)\left[80h^4 T_1^m + 5\gamma^2 h^4 \mathbb{E}|E^m|^2 + 5\gamma h^4 d\right] \\
&= C_1 T_m^1 + 5(1+1/a)\gamma^2 h^4 \mathbb{E}|E^m|^2 + 5(1+1/a)\gamma h^4 d\,,
\end{aligned}
\tag{C.19}
$$

where in the first inequality we use $1 + h^2 < 2$ and

$$C_1 = (1+a)[1 - h/\kappa + Dh^2] + 80(1+1/a)h^4\,.$$

Plug (C.19) in (C.13) and also replace $\mathbb{E}(|E^m|^2)$ with Lemma E.4 equation (E.7), we have

$$
\begin{aligned}
T_1^{m+1} &\leq C_1 T_1^m + \gamma^2\left[10(1+1/a)h^4 + 8h^2\right]\mathbb{E}\left|\widetilde{E}^m\right|^2 \\
&\quad + 100(1+1/a)h^{10}d^2 + 5(1+1/a)\gamma h^4 d + 60h^6 d^2\,,
\end{aligned}
\tag{C.20}
$$

where we use $\gamma L = 1, \eta < h^3$ and $h < 1$.

- **Step 3:** To ensure the decay of $T_1^m$, we need to choose $a$ such that the coefficient in front of $T_1^m$ is strictly smaller than 1. Noting in

$$C_1 = (1+a)[1 - h/\kappa + Dh^2] + 80(1+1/a)h^4$$

the second term is of high order, while the first one is of $1 - O(h)$ amplified by $1 + a$, so it is possible to choose $a$ small enough to make the entire term $1 - O(h)$. Indeed, since $h \leq \frac{1}{(1+D)\kappa}$, we have

$$1 - h/\kappa + Dh^2 \leq 1 - 2h/(3\kappa)\,,$$

and thus by setting $a$ so that

$$1 + a = \frac{1 - h/(2\kappa)}{1 - 2h/(3\kappa)}\,.$$

The entire coefficient is $1 - h/2\kappa + 480\kappa h^3$ and is smaller than 1 for moderately small $h$. Moreover, due to the definition of $a$, we have

$$1 + 1/a \leq 6\kappa/h\,,$$

plugging the calculation in (C.20) we have

$$
\begin{aligned}
T_1^{m+1} &\leq \left\{1 - h/(2\kappa) + 480\kappa h^3\right\}T_1^m \\
&\quad + \gamma^2\left[60\kappa h^3 + 8h^2\right]\mathbb{E}\left|\widetilde{E}^m\right|^2 \\
&\quad + 600\kappa h^9 d^2 + 30\gamma\kappa h^3 d + 60h^6 d^2\,.
\end{aligned}
\tag{C.21}
$$

We further bound $\mathbb{E}|\tilde{E}^m|^2$ by plugging in Lemma E.4 equation (E.6) and use $\gamma L = 1, \kappa h < 1 \leq d, \gamma\kappa = 1/\mu$, we have

$$
\begin{aligned}
T_1^{m+1} &\leq \left\{ 1 - h/(2\kappa) + 480\kappa h^3 \right\} T_1^m \\
&\quad + 84h^2 d\mathbb{E}|\tilde{x}^m - x^m|^2 \\
&\quad + 28\gamma^2 h^2 (24Lh^2 d^4 + 3d\mathbb{E}|\beta^m - \tilde{g}^m|^2) \\
&\quad + 600\kappa h^9 d^2 + 30\gamma\kappa h^3 d + 60h^6 d^2 \qquad , \\
&< \left\{ 1 - h/(2\kappa) + 244h^2 d \right\} T_1^m \\
&\quad + 84\gamma^2 h^2 d\mathbb{E}|\beta^m - \tilde{g}^m|^2 \\
&\quad + 672\gamma h^4 d^4 + 30h^3 d/\mu + 260h^6 d^2
\end{aligned}
\tag{C.22}
$$

where we use $\mathbb{E}|\tilde{x}^m - x^m|^2 \leq \mathbb{E}|\Delta^m|^2 = T_1^m$ and try to absorb small terms into large terms to simplify the formula:

$$
60\kappa h^3 + 8h^2 < 28h^2, \quad 600\kappa h^9 d^2 + 60h^6 d^2 < 260h^6 d^2,
$$

and

$$
480\kappa h^3 + 84h^2 d \leq 244h^2 d, \quad 30\gamma\kappa h^3 d = 30h^3 d/\mu
$$

This proves (C.8).

$\square$

Now we are ready to prove Theorem 5.2 by adjusting $c_p$.

*Proof of Theorem 5.2.* Plug (C.8) and (C.9) into (C.7):

$$
T^{m+1} \leq \left\{ D_1 + \frac{c_p L^2}{d} \right\} T_1^m + \left( 1 - \frac{1}{d} + \frac{D_2}{c_p} \right) c_p T_2^m + D_3 .
$$

Note that according to the definition $D_3$ is of $O(h^3)$, and $D_2$ is of $O(h^2)$ while $D_1 \sim 1 - O(h)$, so it makes sense to choose $c_p$ small enough so that the coefficient for $T_1^m$ keeps being of $1 - O(h)$. Indeed, we let

$$
c_p = 168\gamma^2 h^2 d^2 ,
$$

and will have

$$
\begin{aligned}
T^{m+1} &\leq \left\{ 1 - h/(2\kappa) + 412h^2 d \right\} T_1^m + \left( 1 - \frac{1}{2d} \right) T_2^m , \\
&\quad + 672\gamma h^4 d^4 + 30h^3 d/\mu + 260h^6 d^2
\end{aligned}
\tag{C.23}
$$

where we use $\gamma L = 1$.

Using (17), we can verify

$$
\max\{1 - h/(2\kappa) + 412h^2 d, 1 - 1/2d\} \leq 1 - h/(4\kappa).
$$

Plug into (C.23), we have

$$
T^{m+1} \leq (1 - h/(4\kappa))T^m + 672\gamma h^4 d^4 + 30h^3 d/\mu + 260h^6 d^2 ,
$$

by induction

$$
\begin{aligned}
T^m &\leq (1 - h/(4\kappa))^m T^0 + 2688\gamma\kappa h^3 d^4 + 120\kappa h^2 d/\mu + 1040\kappa h^5 d^2 \\
&\leq (1 - h/(4\kappa))^m T^0 + 2688 h^3 d^4/\mu + 120\kappa h^2 d/\mu + 1040\kappa h^5 d^2 .
\end{aligned}
$$

Finally, consider

$$
\begin{aligned}
T^0 &= \mathbb{E}|\tilde{x}^0 - x^0|^2 + \mathbb{E}|\tilde{w}^0 - \tilde{w}^0|^2 + c_p\mathbb{E}|g^0 - \beta^0|^2 \\
&= \mathbb{E}|\tilde{x}^0 - x^0|^2 + \mathbb{E}|\tilde{w}^0 - \tilde{w}^0|^2 + c_p\mathbb{E}|\nabla f(x^0) - \nabla f(y^0)|^2 \\
&\leq (1 + c_p L^2)(\mathbb{E}|\tilde{x}^0 - x^0|^2 + \mathbb{E}|\tilde{w}^0 - \tilde{w}^0|^2) \leq 2W_2^2(q_0^O, p) ,
\end{aligned}
$$

where we use $168\gamma^2 h^2 d^2 L^2 < 1$. Taking square root on each term and use (C.4), we finally obtain (18). $\square$

# D  Calculation of $\mathbb{E}\left|\widetilde{E}^m\right|^2$ for RCAD-O-LMC

According to the definition of (B.6)-(B.7):

$$\mathbb{E}_{r_m}\widetilde{g}^{m+1} = \widetilde{g}^m + \frac{1}{d}\left(\nabla f(x^m) - \widetilde{g}^m\right), \quad \mathbb{E}_{r_m}\left(\widetilde{g}^{m+1} - \widetilde{g}^m\right) = \frac{1}{d}\left(\nabla f(x^m) - \widetilde{g}^m\right),$$

and

$$\mathbb{E}_{r_m}\left|\widetilde{g}^{m+1} - \widetilde{g}^m\right|^2 = \sum_i \mathbb{E}_{r_m}(\widetilde{g}_i^{m+1} - \widetilde{g}_i^m)^2 = \frac{1}{d}\sum_i |\partial_i f(x^m) - \widetilde{g}_i^m|^2.$$

Naturally

$$\mathbb{E}_{r_m}\widetilde{F}^m = \widetilde{g}^m + (\nabla f(x^m) - \widetilde{g}^m) = \nabla f(x^m).$$

Accordingly,

$$\mathbb{E}_{r_m}\left(\widetilde{E}^m\right) = \nabla f(x^m) - \mathbb{E}_{r_m}(\widetilde{F}^m) = \mathbf{0} \tag{D.1}$$

and

$$\mathbb{E}_{r_m}\left|\widetilde{E}^m\right|^2 = \sum_{i=1}^d \mathbb{E}_{r_m}|\widetilde{E}_i^m|^2 = \sum_{i=1}^d \mathbb{E}_{r_m}\left|\partial_i f(x^m) - \widetilde{g}_i^m - d\left(\widetilde{g}_i^{m+1} - \widetilde{g}_i^m\right)\right|^2. \tag{D.2}$$
$$= (d-1)|\nabla f(x^m) - \widetilde{g}^m|^2.$$

Taking the expectation over the random trajectory:

$$\mathbb{E}\left|\widetilde{E}^m\right|^2 = \mathbb{E}\left(\mathbb{E}_{r_m}|\widetilde{E}^m|^2\right) < d\mathbb{E}|\nabla f(x^m) - \widetilde{g}^m|^2.$$

To analyze each entry of $\partial_i f(x^m) - g_i^m$, we note:

$$|\partial_i f(x^m) - \widetilde{g}_i^m|^2 \le 3\left|\partial_i f(x^m) - \partial_i f(y^m)\right|^2 + 3\left|\partial_i f(y^m) - \beta_i^m\right|^2 + 3\left|\beta_i^m - \widetilde{g}_i^m\right|^2. \tag{D.3}$$

The first term, after taking expectation and summing over $i$, becomes

$$3\mathbb{E}|\nabla f(x^m) - \nabla f(y^m)|^2 \le 3L^2\mathbb{E}|\Delta^m|^2 = 3L^2 T_1^m. \tag{D.4}$$

The last term, with the same procedure, becomes $3T_2^m$. They both will be left in the estimate. We now focus on giving an upper bound of the second term. To do so we adopt a technique from [1, 5]. Define $p = 1/d$, for fixed $m \ge 1$ and $1 \le i \le d$, we have

$$\mathbb{P}(\beta_i^m = \partial_i f(y^0)) = (1-p)^m + (1-p)^{m-1}p$$

and

$$\mathbb{P}(\beta_i^m = \partial_i f(y^j)) = (1-p)^{m-1-j}p, \quad 1 \le j \le m-1$$

$$\mathbb{E}\sum_{i=1}^{d}|\partial_i f(y^m) - \beta_i^m|^2 = \sum_{i=1}^{d}\sum_{j=0}^{m-1}\mathbb{E}(\mathbb{E}(|\partial_i f(y^m) - \beta_i^m|^2|\beta_i^m = \partial_i f(y^j)))\mathbb{P}(\beta_i^m = \partial_i f(y^j))$$

$$= \sum_{j=0}^{m-1}\sum_{i=1}^{d}\mathbb{E}(|\partial_i f(y^m) - \partial_i f(y^j)|^2)\mathbb{P}(\beta_i^m = \partial_i f(y^j))$$

$$\leq^{(I)} \sum_{j=0}^{m-1}\mathbb{E}(|\nabla f(y^m) - \nabla f(y^j)|^2)\mathbb{P}(\beta_1^m = \partial_1 f(y^j))$$

$$\leq L^2 \sum_{j=0}^{m-1}\mathbb{E}(|y^m - y^j|^2)\mathbb{P}(\beta_1^m = \partial_1 f(y^j))$$

$$\leq L^2 \sum_{j=0}^{m-1}\mathbb{E}(|y^m - y^j|^2)(1-p)^{m-1-j}p$$
$$+ L^2\mathbb{E}(|y^m - y^0|^2)(1-p)^m$$

$$\leq^{(II)} L^2\sum_{j=0}^{m-1}\mathbb{E}\left(\left|\int_{jh}^{mh}\nabla f(y_s)ds - \sqrt{2h}\sum_{i=j}^{m-1}\xi_i\right|^2\right)(1-p)^{m-1-j}p$$

$$+ L^2\mathbb{E}\left(\left|\int_{0}^{mh}\nabla f(y_s)ds - \sqrt{2h}\sum_{i=0}^{m-1}\xi_i\right|^2\right)(1-p)^m$$

$$\leq^{(III)} L^2\sum_{j=0}^{m-1}\left[2h^2(m-j)^2\mathbb{E}_p|\nabla f(y)|^2 + 4hd(m-j)\right](1-p)^{m-1-j}p$$

$$+ L^2\left[2h^2m^2\mathbb{E}_p|\nabla f(y)|^2 + 4hdm\right](1-p)^m$$

$$\leq^{(IV)} 2ph^2L^2\mathbb{E}_p|\nabla f(y)|^2\left[\sum_{j=1}^{m}j^2(1-p)^{j-1} + m^2(1-p)^m/p\right]$$

$$+ 4phL^2d\left[\sum_{j=1}^{m}j(1-p)^{j-1} + m(1-p)^m/p\right]$$

$$\leq^{(V)} \frac{8h^2L^2\mathbb{E}_p|\nabla f(y)|^2}{p^2} + \frac{8hL^2d}{p}$$

$$\leq^{(VI)} 8hL^2d^2\left[\frac{hL^2d}{\mu} + 1\right]$$

(D.5)

where in (I) we use $\mathbb{P}(\beta_i^m = \partial_i f(y^j))$ are same for different $i$, (II) comes from (B.4),(B.5), (III) comes from $y_t \sim p$ for any $t$, (IV) comes from changing of variable, in (V) we use the bound for terms in the bracket and in (VI) we use $\mathbb{E}_p|x - x^*|^2 \leq d/\mu$ according to Theorem D.1 in [1], where $x^*$ is the maximum point of $f$.

In conclusion, we have

$$\mathbb{E}\left|\widetilde{E}^m\right|^2 \leq 3dL^2T_1^m + 3dT_2^m + 24hL^2d^3\left[\frac{hL^2d}{\mu} + 1\right].$$

(D.6)

# E   Key lemma in proof of RCAD-U-LMC

**Lemma E.1.** *Under conditions of Theorem 5.2, $\left(\widetilde{X}_t, \widetilde{V}_t\right)$ are defined in* (C.5)*, we have*

$$\mathbb{E}\int_{mh}^{(m+1)h}\left|\widetilde{X}_t - \widetilde{x}^m\right|^2 \mathrm{d}t \leq \frac{h^3\gamma d}{3} \tag{E.1}$$

*and*

$$\mathbb{E}\int_{mh}^{(m+1)h}\left|\left(\widetilde{V}_t - V_t\right) - (\widetilde{v}^m - v^m)\right|^2 \mathrm{d}t \leq 16h^3\mathbb{E}|\Delta^m|^2 + \gamma^2 h^3\mathbb{E}|E^m|^2 + 0.4\gamma h^5 d\,, \tag{E.2}$$

**Lemma E.2.** *Under conditions of Theorem 5.2, and $B^m, D^m$ are defined in* (C.15),(C.17)*, we have*

$$\mathbb{E}|B^m|^2 \leq 32h^4\mathbb{E}|\Delta^m|^2 + 2\gamma^2 h^4\mathbb{E}|E^m|^2 + 2\gamma h^4 d \tag{E.3}$$

$$\mathbb{E}|D^m|^2 \leq 16h^4\mathbb{E}|\Delta^m|^2 + \gamma^2 h^4\mathbb{E}|E^m|^2 + 0.4\gamma h^6 d \tag{E.4}$$

**Lemma E.3.** *Under conditions of Theorem 5.2, and $A^m, C^m$ defined in* (C.14),(C.16)*, there exists a uniform constant $D$ such that*

$$\mathbb{E}((1+h^2)|A^m|^2 + |C^m|^2) \leq \left[1 - h/\kappa + Dh^2\right]\mathbb{E}|\Delta^m|^2 \tag{E.5}$$

*where $\kappa = L/\mu$ is the condition number of $f$.*

**Lemma E.4.** *Under conditions of Theorem 5.2, we have estimation for approximation gradient*

$$\mathbb{E}|\widetilde{E}^m|^2 \leq 3dL^2\mathbb{E}|\widetilde{x}^m - x^m|^2 + 24Lh^2 d^4 + 3d\mathbb{E}\,|\beta^m - \widetilde{g}^m|^2 \tag{E.6}$$

*and*

$$\mathbb{E}|E^m|^2 \leq 2\mathbb{E}|\widetilde{E}^m|^2 + 20L^2 h^6 d^2\,. \tag{E.7}$$

We prove these four lemmas below.

*Proof of Lemma E.1.* First we prove (E.1). According to (C.5), we have

$$\begin{aligned}
\mathbb{E}\int_{mh}^{(m+1)h}\left|\widetilde{X}_t - \widetilde{x}^m\right|^2 dt &= \mathbb{E}\int_{mh}^{(m+1)h}\left|\int_{mh}^t \widetilde{V}_s ds\right|^2 dt \\
&\leq \int_{mh}^{(m+1)h}(t-mh)\int_{mh}^t \mathbb{E}\left|\widetilde{V}_s\right|^2 ds dt \\
&= \int |v|^2 p_2(x,v)\,\mathrm{d}x\,\mathrm{d}v\int_{mh}^{(m+1)h}(t-mh)^2 dt = \frac{h^3\gamma d}{3}\,,
\end{aligned} \tag{E.8}$$

where in the first inequality we use Hölder's inequality, and for the second equality we use $p_2$ is a stationary distribution so that $\left(\widetilde{X}_t, \widetilde{V}_t\right) \sim p_2$ and $\widetilde{V}_t \sim \exp(-|v|^2/(2\gamma))$ for any $t$.

Second, to prove (E.2), using (C.2),(C.5), we first rewrite $\left(\widetilde{V}_t - V_t\right) - (\widetilde{v}^m - v^m)$ as

$$\begin{aligned}
\left(\widetilde{V}_t - V_t\right) - (\widetilde{v}^m - v^m) &= (\widetilde{v}^m - v^m)\left(e^{-2(t-mh)} - 1\right) \\
&\quad - \gamma\int_{mh}^t e^{-2(t-s)}\left[\nabla f(\widetilde{X}_s) - \nabla f(x^m)\right]\mathrm{d}s \\
&\quad + \gamma\int_{mh}^t e^{-2(t-s)}\,\mathrm{d}s E^m \\
&= \mathrm{I}(t) + \mathrm{II}(t) + \mathrm{III}(t)\,.
\end{aligned} \tag{E.9}$$

for $mh \leq t \leq (m+1)h$. Then we bound each term seperately:

- 

$$\mathbb{E} \int_{mh}^{(m+1)h} |\mathrm{I}(t)|^2 \, \mathrm{d}t \le h\mathbb{E} \int_{mh}^{(m+1)h} \left| (\widetilde{v}^m - v^m)\left(e^{-2(t-mh)} - 1\right) \right|^2 \, \mathrm{d}t$$

$$\le h \int_{mh}^{(m+1)h} (2(t-mh))^2 \mathbb{E} \left| \widetilde{v}^m - v^m \right|^2 \, \mathrm{d}t \qquad (\text{E.10})$$

$$\le \frac{4h^3}{3} \mathbb{E} \left| \widetilde{v}^m - v^m \right|^2 \, ,$$

where we use Hölder's inequality in the first inequality and $1 - e^{-x} < x$ in the second inequality.

- 

$$\mathbb{E} \int_{mh}^{(m+1)h} |\mathrm{II}(t)|^2 \, \mathrm{d}t \le \gamma^2 \mathbb{E} \int_{mh}^{(m+1)h} \left| \int_{mh}^t e^{-2(t-s)} \left[ \nabla f(\widetilde{\mathrm{X}}_s) - \nabla f(x^m) \right] \mathrm{d}s \right|^2 \, \mathrm{d}t$$

$$\le 2\gamma^2 \mathbb{E} \int_{mh}^{(m+1)h} \left| \int_{mh}^t e^{-2(t-s)} \left[ \nabla f(\widetilde{\mathrm{X}}_s) - \nabla f(\widetilde{x}^m) \right] \mathrm{d}s \right|^2 \, \mathrm{d}t$$

$$+ 2\gamma^2 \mathbb{E} \int_{mh}^{(m+1)h} \left| \int_{mh}^t e^{-2(t-s)} \left[ \nabla f(\widetilde{x}^m) - \nabla f(x^m) \right] \mathrm{d}s \right|^2 \, \mathrm{d}t$$

$$\le 2\gamma^2 \int_{mh}^{(m+1)h} (t-mh)\mathbb{E} \int_{mh}^t \left| \nabla f(\widetilde{\mathrm{X}}_s) - \nabla f(\widetilde{x}^m) \right|^2 \, \mathrm{d}s \, \mathrm{d}t$$

$$+ 2\gamma^2 \int_{mh}^{(m+1)h} (t-mh)\mathbb{E} \int_{mh}^t \left| \nabla f(\widetilde{x}^m) - \nabla f(x^m) \right|^2 \, \mathrm{d}s \, \mathrm{d}t$$

$$\le 2\gamma^2 L^2 \int_{mh}^{(m+1)h} (t-mh)\mathbb{E} \int_{mh}^t \left| \widetilde{\mathrm{X}}_s - \widetilde{x}^m \right|^2 \, \mathrm{d}s \, \mathrm{d}t$$

$$+ 2\gamma^2 L^2 \int_{mh}^{(m+1)h} (t-mh)\mathbb{E} \int_{mh}^t |\widetilde{x}^m - x^m|^2 \, \mathrm{d}s \, \mathrm{d}t$$

$$\le 2\gamma^3 L^2 d \int_{mh}^{(m+1)h} \frac{(t-mh)^4}{3} \, \mathrm{d}t + 2\gamma^2 L^2 \int_{mh}^{(m+1)h} (t-mh)^2 \, \mathrm{d}t \mathbb{E} |\widetilde{x}^m - x^m|^2$$

$$\le \frac{2\gamma^3 L^2 h^5 d}{15} + \frac{2\gamma^2 L^2 h^3}{3} \mathbb{E} |\widetilde{x}^m - x^m|^2 \, ,$$

(E.11)

where in the third inequality we use gradient of $f$ is $L$-Lipschitz function and we use (E.1) in the fourth inequality.

- 

$$\mathbb{E} \int_{mh}^{(m+1)h} |\mathrm{III}(t)|^2 \, \mathrm{d}t = \gamma^2 \mathbb{E} \int_{mh}^{(m+1)h} \left| \int_{mh}^t e^{-2(t-s)} \, \mathrm{d}s E^m \right|^2 \, \mathrm{d}t$$

$$\le \gamma^2 \int_{mh}^{(m+1)h} (t-mh)^2 \, \mathrm{d}t \mathbb{E}(|E^m|^2) \qquad (\text{E.12})$$

$$\le \frac{\gamma^2 h^3}{3} \mathbb{E}(|E^m|^2) \, ,$$

Plug (E.10),(E.11),(E.12) into (E.9) and using $\gamma L = 1$, we have

$$\mathbb{E} \int_{mh}^{(m+1)h} \left| \left( \widetilde{\mathrm{V}}_t - \mathrm{V}_t \right) - (\widetilde{v}^m - v^m) \right|^2 \, \mathrm{d}t$$

$$\le 3 \left( \mathbb{E} \int_{mh}^{(m+1)h} |\mathrm{I}(t)|^2 \, \mathrm{d}t + \mathbb{E} \int_{mh}^{(m+1)h} |\mathrm{II}(t)|^2 \, \mathrm{d}t + \mathbb{E} \int_{mh}^{(m+1)h} |\mathrm{III}(t)|^2 \, \mathrm{d}t \right)$$

$$\le 4h^3 \left( \mathbb{E} |\widetilde{x}^m - x^m|^2 + \mathbb{E} |\widetilde{v}^m - v^m|^2 \right) + \gamma^2 h^3 \mathbb{E}(|E^m|^2) + 0.4\gamma h^5 d \, ,$$

using (C.3), we get the desired result. $\qquad \square$

*Proof of Lemma E.2.* First, we seperate $B^m$ into two parts:

$$\mathbb{E}|B^m|^2 \leq 2\mathbb{E}\left|\int_{mh}^{(m+1)h}\left(\widetilde{V}_t - V_t\right) - (\widetilde{v}^m - v^m)\,\mathrm{d}t\right|^2$$

$$+ 2\mathbb{E}\left|\gamma\int_{mh}^{(m+1)h}e^{-2((m+1)h-t)}\left[\nabla f(\widetilde{X}_t) - \nabla f(\widetilde{x}^m)\right]\,\mathrm{d}t\right|^2.$$

And each terms can be bounded:

- $$\mathbb{E}\left|\int_{mh}^{(m+1)h}\left(\widetilde{V}_t - V_t\right) - (\widetilde{v}^m - v^m)\,\mathrm{d}t\right|^2$$
$$\leq h\mathbb{E}\int_{mh}^{(m+1)h}\left|\left(\widetilde{V}_t - V_t\right) - (\widetilde{v}^m - v^m)\right|^2\,\mathrm{d}t \tag{E.13}$$
$$\leq 16h^4\mathbb{E}|\Delta^m|^2 + \gamma^2 h^4\mathbb{E}(|E^m|^2) + 0.4\gamma h^6 d\,,$$

  where we use Lemma E.1 (E.2) in the second inequality.

- $$\mathbb{E}\left|\gamma\int_{mh}^{(m+1)h}e^{-2((m+1)h-t)}\left[\nabla f(\widetilde{X}_t) - \nabla f(\widetilde{x}^m)\right]\,\mathrm{d}t\right|^2$$
$$\leq h\gamma^2\mathbb{E}\int_{mh}^{(m+1)h}\left|e^{-2((m+1)h-t)}\left[\nabla f(\widetilde{X}_t) - \nabla f(\widetilde{x}^m)\right]\right|^2\,\mathrm{d}t \tag{E.14}$$
$$\leq h\gamma^2 L^2\mathbb{E}\int_{mh}^{(m+1)h}\left|\widetilde{X}_t - \widetilde{x}^m\right|^2\,\mathrm{d}t$$
$$\leq \frac{h^4\gamma^3 L^2 d}{3} \leq \frac{h^4\gamma d}{3}\,,$$

  where we use Lemma E.1 (E.1) and $\gamma L = 1$ in the last two inequalities.

Combine (E.13),(E.14) together, we finally have

$$\mathbb{E}|B|^2 \leq 32h^4\mathbb{E}|\Delta^m|^2 + 2\gamma^2 h^4\mathbb{E}(|E^m|^2) + 0.8h^6\gamma d + 2h^4\gamma d/3\,,$$

which implies (E.3) if we further use $h < 1$.

Next, estimation of $\left(\mathbb{E}|D|^2\right)^{1/2}$ is a direct result of (E.13). $\qquad\square$

*Proof of Lemma E.3.* Let $\widetilde{x}^m - x^m = a$ and $\widetilde{w}^m - w^m = b$. First, by the mean-value theorem, there exists a matrix $H$ such that $\mu I_d \preceq H \preceq L I_d$ and

$$\nabla f(\widetilde{x}^m) - \nabla f(x^m) = Ha\,.$$

By calculation, $\int_{mh}^{(m+1)h}e^{-2((m+1)h-t)}\,\mathrm{d}t = \frac{1-e^{-2h}}{2}$ and

$$A^m = (h + e^{-2h})(\widetilde{v}^m - v^m) + \left(I_d - \frac{(1-e^{-2h})}{2}\gamma H\right)(\widetilde{x}^m - x^m)$$
$$= \left(\left(1 - h - e^{-2h}\right)I_d - \frac{(1-e^{-2h})}{2}\gamma H\right)a + (h + e^{-2h})b$$

$$C^m = (1-h)a + hb\,.$$

Since $\|\gamma H\|_2 \leq 1$ and we also have following calculation

$$h + e^{-2h} = h + e^{-2h} - 1 + 1 = 1 - h + O(h^2)\,,$$
$$1 - h - e^{-2h} = h + O(h^2)\,,$$

$$1 - e^{-2h} = 2h + O(h^2) \,.$$

If we further define matrix $\mathcal{M}_A$ and $\mathcal{M}_C$ such that

$$|A^m|^2 = (a,b)^\top \mathcal{M}_A (a,b) \,, \quad |C^m|^2 = (a,b)^\top \mathcal{M}_C (a,b) \,,$$

then, we have

$$\left\| \mathcal{M}_A - \begin{bmatrix} 0 & hI_d - \gamma hH \\ hI_d - \gamma hH & (1-2h)I_d \end{bmatrix} \right\|_2 \le D_1 h^2 \,,$$

and

$$\left\| \mathcal{M}_B - \begin{bmatrix} (1-2h)I_d & hI_d \\ hI_d & 0 \end{bmatrix} \right\|_2 \le D_1 h^2 \,,$$

where $D_1$ is a uniform constant since $h < 1/1648$ by (17). This further implies

$$(1+h^2)|A^m|^2 + |C^m|^2 = (a,b)^\top \begin{bmatrix} (1-2h)I_d & 2hI_d - \gamma hH \\ 2hI_d - \gamma hH & (1-2h)I_d \end{bmatrix} (a,b) + h^2 (a,b)^\top Q (a,b)$$

where $\|Q\|_2 \le D_2$ and $D_2$ is a uniform constant. Calculate the eigenvalue of the dominating matrix (first term), we need to solve

$$\det \left\{ (1 - 2h - \lambda)^2 I_d - (2hI_d - \gamma hH)^2 \right\} = 0 \,,$$

which implies eigenvalues $\{\lambda_j\}_{j=1}^d$ solve

$$(1 - 2h - \lambda_j)^2 - (2h - \gamma h \Lambda_j)^2 = 0 \,,$$

where $\Lambda_j$ is $j$-th eigenvalue of $H$. Since $\gamma \Lambda_j \le \gamma L = 1$ and $h < 1$, we have

$$\lambda_j \le 1 - \gamma \Lambda_j h \le 1 - \mu h \gamma = 1 - h/\kappa$$

for each $j = 1, \ldots, d$. This implies

$$\left\| \begin{bmatrix} (1-2h)I_d & 2hI_d - \gamma hH \\ 2hI_d - \gamma hH & (1-2h)I_d \end{bmatrix} \right\|_2 \le 1 - h/\kappa \,,$$

and

$$(1+h^2)|A^m|^2 + |C^m|^2 \le (1 - h/\kappa + Dh^2)(|a|^2 + |b|^2) \,,$$

where $D$ is a uniform constant. Take expectation on both sides, we obtain (E.5).

$\square$

*Proof of Lemma E.4.* The proof is mostly the same as that in the calculation in Appendix D. Inequality (D.3) still holds true except the second term needs to be treated differently. Following the step in Appendix D, we define $p = 1/d$, and then for fixed $m \ge 1$ and $1 \le i \le d$, we have

$$\mathbb{P}(\beta_i^m = \partial_i f(\widetilde{x}^0)) = (1-p)^m + (1-p)^{m-1}p \,,$$

and

$$\mathbb{P}(\beta_i^m = \partial_i f(\widetilde{x}^j)) = (1-p)^{m-1-j}p, \quad 1 \le j \le m-1 \,.$$

$$\mathbb{E}\sum_{i=1}^{d}|\partial_i f(\widetilde{x}^m) - \beta_i^m|^2 = \sum_{i=1}^{d}\sum_{j=0}^{m-1}\mathbb{E}(\mathbb{E}(|\partial_i f(\widetilde{x}^m) - \beta_i^m|^2|\beta_i^m = \partial_i f(\widetilde{x}^j)))\mathbb{P}(\beta_i^m = \partial_i f(\widetilde{x}^j))$$

$$\leq \sum_{j=0}^{m-1}\sum_{i=1}^{d}\mathbb{E}(|\partial_i f(\widetilde{x}^m) - \partial_i f(\widetilde{x}^j)|^2)\mathbb{P}(\beta_i^m = \partial_i f(\widetilde{x}^j))$$

$$\leq \sum_{j=0}^{m-1}\mathbb{E}(|\nabla f(\widetilde{x}^m) - \nabla f(\widetilde{x}^j)|^2)\mathbb{P}(\beta_1^m = \partial_1 f(\widetilde{x}^j))$$

$$\leq L^2 \sum_{j=0}^{m-1}\mathbb{E}(|\widetilde{x}^m - \widetilde{x}^j|^2)\mathbb{P}(\beta_1^m = \partial_1 f(\widetilde{x}^j))$$

$$\leq L^2 \sum_{j=0}^{m-1}\mathbb{E}(|\widetilde{x}^m - \widetilde{x}^j|^2)(1-p)^{m-1-j}p$$
$$+ L^2\mathbb{E}(|\widetilde{x}^m - \widetilde{x}^0|^2)(1-p)^m$$

$$\leq^{(II)} L^2 \sum_{j=0}^{m-1}\mathbb{E}\left(\left|\int_{jh}^{mh}\widetilde{V}_s ds\right|^2\right)(1-p)^{m-1-j}p$$
$$+ L^2\mathbb{E}\left(\left|\int_{0}^{mh}\widetilde{V}_s ds\right|^2\right)(1-p)^m$$

$$\leq^{(III)} L^2 \sum_{j=0}^{m-1}\left[2h^2(m-j)^2\mathbb{E}_{p_2}|\widetilde{V}|^2\right](1-p)^{m-1-j}p$$
$$+ L^2\left[2h^2 m^2 \mathbb{E}_{p_2}|\widetilde{V}|^2\right](1-p)^m$$

$$\leq^{(IV)} 2ph^2 L^2 \mathbb{E}_{p_2}|\widetilde{V}|^2 \left[\sum_{j=1}^{m}j^2(1-p)^{j-1} + m^2(1-p)^m/p\right]$$

$$\leq^{(V)} \frac{8h^2 L^2 \mathbb{E}_{p_2}|\widetilde{V}|^2}{p^2}$$
$$\leq^{(VI)} 8\gamma h^2 L^2 d^3 = 8h^2 L d^3 \,,$$

(E.15)

where (II) comes from (C.5), (III) comes from $\left(\widetilde{X}_t, \widetilde{V}_t\right) \sim p_2$ for any $t$, (IV) comes from changing of variable, in (V) we use the bound for terms in the bracket and in (VI) we use $\mathbb{E}_{p_2}|v|^2 \leq \gamma d$. This inequality differ from the derivation in Appendix D only through (II).

Next, to prove (E.7), we only need to notice

$$\mathbb{E}|E^m|^2 \leq 2\mathbb{E}|\widetilde{E}^m|^2 + 2\mathbb{E}|F^m - \widetilde{F}^m|^2 \,,$$

(B.9) and $\eta < h^3$ and follow the same calculation as in done in Appendix D. $\qquad\square$

## Footnotes

[1]In particular, it is obvious that the square of all terms except $\Delta^m$ contribute small values and will enter $d$, and the cross terms would dominate.