[Reviews · NeurIPS 2020]

Review 1

Summary and Contributions: The paper proposes to integrate Random Coordinate Descent (RCD) into Langevin Monte Carlo (LMC)(both overdamped and underdamped version) framework and theoretically analyzes its convergence properties. The paper presents one negative result when directly apply RCD in overdamped LMC.

Strengths: The idea presented is well motivated and clearly presented. And the work may be useful for wide range of applications, especially in the case when gradient info. is not readily available. It may also have potential in generative modeling area.

Weaknesses: This paper is mainly a theoretical paper with convergence analysis included, however, it would not be so convincing without comparisons on toy data. Therefore, it would be better to also numerically and empirically compare with the traditional LMC in terms accuracy and computation efficiency on the synthetic/simulated data. There is a gap between theoretical analysis and empirically performance in general, without illustration on simplest toy data might not be easy to let general audience to embrace the usefulness of the proposed algorithm.

Correctness: The algorithm/methodology seems correct. The convergence proof correctness is beyond my scope and thus unjudgeable.

Clarity: The paper is well written.

Relation to Prior Work: Yes

Reproducibility: Yes

Additional Feedback: The author rebuttal did address some of my concerns, more illustrative simulated experiments could be included for demonstration in their revision.


Review 2

Summary and Contributions: I have read the other reviews and author feedback carefully. It would be great if the authors could add the simulation figures in the final paper. ------ This paper considers the problem of sampling from a strongly log-concave distribution when gradient is not available and needs to be approximated via finite-difference. The naive implementation of the randomized coordinate descent does not improved upon the direct finite difference approximation of gradients. The authors propose a new variance reduction approach to implement the randomized coordinate averaging descent (RCAD) to be combined with Langevin dynamics for log-concave sampling. With K denoting dimension, Langevin Monte Carlo (LMC) with direct finite difference approximation of gradients converges in K^2/epsilon function evaluations because each step computes K coordinates of the gradient. The proposed RCAD-O-LMC converges in K^{3/2}/epsilon function evaluation. Similar improvement of RCAD-U-LMC (the underdamped Langevin with RCAD) over U-LMC (underdamped Langevin) is also derived. The improvement is largely inspired by the variance reduction techniques for stochastic gradient methods on a finite sum function in optimization.

Strengths: In the specific setting of strongly log-concave sampling where gradients need to be approximated via finite difference, their methods have improved dimension dependency K on the number of function evaluations. The proposed algorithm is not that novel for optimization, but for sampling, the proposed algorithm is new.

Weaknesses: 1. Some parts of the paper writing seems to be rushed and contains small errors. For example, in the main Theorem 5.1 equation (16) (line 238), it should be \sqrt{1 + R^2} instead of \sqrt{1 + 1/R^2}. Because According to line 108 of Appendix, due to "c_p" choice, R^2 should appear there. It is a mistake. There are also other typos in the Appendix that I will explain below. 2. This paper does not have a convincing application case for their theoretical setting. For the proposed algorithms to have improvement, one has to be in a setting where gradient computation is much more expensive than function evaluation. Could the authors provide a good example? For statistical problems like Bayesian logistic regression, the computation cost for the gradient is similar to function evaluation. If one has a neural network, then the forward cost (function evaluation) and the backward cost (gradient evaluation) have the same order of computation. 3. It would be better if the authors provide some simulation results to check the suggested rate improvement. 4. Theorem 5.1 requires Hessian-Lipschitz condition. Is it possible to provide a version without Hessian-Lipschitz condition in the main paper? 5. The proposed technique seems only work for Wasserstein-2 distance, can the authors comment on other distributional distance?

Correctness: I have checked the proof of Theorem 5.1. The main proof is correct, except for some small typos. For example, Appendix line 108, line 100 1-1/K instead of 1-K, line 103 + is missing etc.

Clarity: The main paper is clearly written. Maybe the heavy notation in Theorem 5.1 and Theorem 5.2 can be cleaned up a bit. Big O notation in Table 1 is not well defined. What is being ignored in big O notation? just constant, or also the condition number? The Appendix needs more work. There are many typos to be fixed. The way the current proof is written is not easy to read. Adding a paragraph or a diagram in the begining of Appendix B to explain the proof sketch might be helpful. Inconsistent notation, like putting subscript 2 in the l2 norm in Line 73 of Appendix should be avoided. Letter B is overloaded, for Brownian motion and for the constant.

Relation to Prior Work: Yes. To my knowledge, this specific type of log-concave sampling where gradient needs to be approximated via function evaluation is not studied before.

Reproducibility: Yes

Additional Feedback: The title could be more clear about the specific setting where gradient needs to be approximated by finite difference. It is not just about high-dimension.


Review 3

Summary and Contributions: The paper deals with the problem of efficient sampling from log-concave distributions in a high dimensional setting. In this setup, a simple yet efficient sampling algorithms is the Langevin Monte Carlo algorithm. It makes use of the gradient of the (log-)pdf to ensure a fast convergence towards the region of high density. However, computing the gradient in a high-dimensional setting is computationally expensive. The contributions of the paper are related to techniques for reducing the computational cost for evaluating the gradient, and are two-fold: first, an analysis of the simple random coordinate descent (RCD) technique for reducing the computational cost for the gradient is carried out, and second, a new technique to reduce the computational cost for gradient evaluation is proposed and analysed. The contributions are theoretical. NOTE: I read the authors rebuttal.

Strengths: Overall, the paper proposes an interesting technique for reducing the per-iteration computational cost of gradient evaluation while still maintaing an overall acceptable computational cost. The technique is employed in two sampling algorithms. The paper starts by analysing an existing technique for reducing the per-iteration computational cost of the gradient evaluation in sampling algorithms, namely random coordinate descent (RCD). The analysis is carried out in the general case and the result is that in such a setting for a given required accuracy there is no overall net gain in computational cost as the per-iteration reduced cost is balanced out by the increase in the required number of iterations. Such a theoretical analysis is interesting and the results are, in my opinion, noteworthy. The authors also point out that there exist particular scenarios in which the RCD approach yields an overall net gain in computational cost for a given accuracy. The paper then proposes an improved technique for reducing the per-iteration computational cost for gradient evaluation. It is important to note that the 'new' gradient that is computed at each iteration and that has a reduced computational cost is in fact an approximation of the true gradient. The issue with the RCD technique is that the approximation has a rather high variance, hence the need for a large number of iterations to reach the required accuracy. The proposed technique starts with a complete gradient evaluation and then at each iteration proceeds to update the gradient in only one direction, which is chosen at random. For the other directions, the technique calls for recycling the gradients from the previous iteration irrespective if they have been updated or not. Theoretical results for the distance between the distribution of the iterates at iteration m and the target distribution are given for both algorithms that are considered in the paper. The improved technique for approximating the gradient and the theoretical results for the distance between the distribution of the iterates and the target distribution are the strong points of the paper. From my point of view, these contributions are of interest for the ML/NeurIPS community.

Weaknesses: The main weaknesse of the paper is that there are no numerical examples of the proposed technique and algorithms. I feel that the authors devoted too much space explaining the existing algorithms: the contributions are discussed starting with section 4 which starts at the bottom of the 5th page. Space could have been easily gained to make room for a section containing a numerical evaluation of the performances of the proposed technique and algorithms. If we look at theorems 5.1 and 5.2 we see that there is a rather intricate dependence of the theoretical results with respect to what we can collectively refer to as parameters of the proposed methods (h, \eta, M, L), the number of iterations and the dimension of input data. A numerical evaluation would have definitely allowed one to gain some insight on the performances of the proposed methods for real-life problems, more so as the contributions are given in a general setting. I am left wondering why did one not consider the case of stochastic optimization and focused solely on sampling. The technique that is used to reduce the number of gradient evaluations in one iteration could be easily applied to an optimization problem. Personally, I would have first studied the advantages offered by the proposed technique for an optimization problem. The fact that the paper analyses the proposed technique in a general setting is a strong point. However, nothing is said about particular cases. How does the proposed technique fares with RCD in such a case, especially in a case in which RCD offers good results? Also, nothing is said about how the method fares with other competing methods besides LMC and RCD.

Correctness: The claims seem to be correct. There are no numerical experiments.

Clarity: The paper is mostly clearly written, though some paragraphs from the introduction section are somewhat ambiguous, namely the last paragraph on page 1 and the first two paragraphs on page 2: - In the last paragraph on page 1 one contrasts the broad class of MCMC algorithms with particular instances of MCMC type algorithms: MH algorithm, Langevin dynamics based methods, Hamiltonian Monte Carlo, etc. There is no point in contrasting a class of algorithms with particular instances of algorithms. - In the first paragraph on page 2 one says that MCMC is one popular method. I would not call MCMC a method, MCMC is a class of algorithms. Furthermore, from reading the first and second paragraph on page 2 I get the impression that Langevin dynamics is not an MCMC type algorithm, when it is the case.

Relation to Prior Work: The relation to prior work is in my opinion clearly discussed.

Reproducibility: Yes

Additional Feedback:

[Author Response · NeurIPS 2020]

We thank reviewers for careful reading, and appreciate the honesty of Reviewer 2. The weakness pointed out here is the lack of numerical evidence, which we explain below (No. 2). As both other reviewers pointed out, the significance of the paper is two-fold: 1. We show, by giving a counter-example, that the direct application of RCD (random coordinate descent) on LMC (Langevin Monte Carlo) does not improve the numerical performance; 2. With variance reduction techniques incorporated, the numerical cost is significantly reduced. The reduction rate depends on the dimension of the problem: the algorithm saves more in high dimensional problems. Moreover, in the under-damped case, the method converges as fast as the vanilla LMC while requiring only one partial derivative instead of the full gradient per iteration. This is the optimal numerical cost one can possibly get. Below we address the weakness pointed out by the reviewers.

1. WHAT IF THE FORWARD COST (FUNCTION EVALUATION) AND THE BACKWARD COST (GRADIENT EVALUATION) HAVE THE SAME ORDER OF COMPUTATION? We agree that there are cases, as pointed out by Reviewer 3 when the two costs are similar, but in the most general setting, a problem does require a much higher cost for the gradient to be computed. In fact, most problems in atmospheric science and remote sensing cannot even have one gradient computed due to the high dimensionality (see Refs. 21, 36). This is exactly why the ensemble type sampling methods became popular that target at achieving "gradient-free" property. We would like to put ourselves in the most general footing. It is our principle, and we believe it is shared by most researchers, that investigation into special cases should come after a clear picture of general setups. The same question could have been asked to challenge the validity of RCD, but nevertheless RCD is a tremendously popular method in optimization. We agree with Reviewer 4 that we could have made some comments on the cases when RCD already performs well. We believe playing with directional Lipschitz constants would be the key but this is beyond the scope of the current paper.

2. WHY ARE THERE NO NUMERICAL RESULTS? We have not seen a single result in the literature, including the fundamental papers in the area (see Refs. 8, 10, 12), that truly demonstrates the convergence in Wasserstein-2 distance numerically. This is simply because there is no numerical method available yet that is even able to evaluate the criterion. The $W_2$ distance between two probability measures is hard to compute in high dimensions, especially when one probability is represented by one data point. In the plot below we show the decay of MSE (mean square error, a much weaker and loosened criterion). We could have chosen to demonstrate these in the original paper, but we preferred reserving the space for richer theoretical guarantees other than providing numerical results with mismatching norms.

Figure 1: Decay of MSE of LMC in overdamped (left) and underdamped (right) settings. Test function: $\phi(x) = x_1^2$.

3. CAN WE ELIMINATE THE HESSIAN-LIPSCHITZ CONTINUITY ASSUMPTION? Yes we can. We did comment on it in the original paper (line 245-247). Since neither the result nor the proof is significant, we did not include the full statement in the paper. **Theorem:** *Suppose $f$ satisfies Assumption 3.1, $h < O\left(\frac{1}{K}\right)$ and $\eta < h$, then:* $W_2(q_m^O, p) \lesssim \exp(-Mhm/4)W_2(q_0^O, p) + hK^{3/2} + h^{1/2}K$. Here $\lesssim$ means $\leq$ up to a constant independent of $h, K$.

4. CAN WE CHANGE OUR NORM? We can comment on other norms but we do not believe any theorems on other norms should be included. There is no single paper (either journal or conference) in the literature that studies convergence in more than one norm in one paper, exactly because different criteria are evaluated with different mathematical techniques, and the entire roadmap has to change. We do have a very simple corollary on MSE convergence. It is a standard derivation from $W_2$ convergence. **Corollary:** *Under conditions of Theorem 5.1, MSE decays with the rate* $|\mathbb{E}_{q_m^O}(\phi) - \mathbb{E}_p(\phi)| \lesssim \exp(-Mhm/4)W_2(q_0^O, p) + h(K^{3/2} + K)$, *for all Lipschitz test function $\phi$.*

5. (FROM REVIEWER 4) WHY DON'T WE DO OPTIMIZATION FIRST? We very much appreciate Reviewer 4 raising this question. The fact is, we did. It was a surprising result for us that in optimization, this formulation does **NOT** help in saving numerical cost. We were left wondering if this is a known result in the community that we missed out on, or is this also new? We opt to investigate it in the optimization setup a bit more before claiming publicly a negative result.

Finally, while we agree a different layout of the paper, and some changes in phrases may help delivering stronger messages to some certain audience, and small typos on constants should also have been avoided, we are genuinely surprised that some tasks that have never been done in the literature are used to discount the significance of the current paper. We will be happily corrected by the reviewers if we miss any older results, and we will continue monitoring the area and NeurIPS selected publications for most recent progresses. We do not think the paper has ethical impact.

[Meta-Review · NeurIPS 2020]

The reviewers are in consensus that there is value in this paper. We (the AC and SAC) have read and discussed the submission. We urge the authors to carefully consider the reviewers' comments, which could significantly increase the impact of the paper. In addition, the paper could be improved from a clarity perspective. Finally, we follow the reviewers in urging that some numerical examples be added, of the kind in the author response. These serve a critical function for readers in demonstrating the phenomena being studied, which makes things much more concrete for readers.